# Decentralized, Communication- and Coordination-free Learning in Structured Matching Markets

**Chinmay Maheshwari**
EECS
University of California Berkeley
chinmay_maheshwari@berkeley.edu

**Shankar Sastry**
EECS
University of California Berkeley
sastry@eecs.berkeley.edu

**Eric Mazumdar**
CMS and Economics
Caltech
mazumdar@caltech.edu

## Abstract

We study the problem of online learning in competitive settings in the context of two-sided matching markets. In particular, one side of the market, the agents, must learn about their preferences over the other side, the firms, through repeated interaction while competing with other agents for successful matches. We propose a class of decentralized, communication- and coordination-free algorithms that agents can use to reach to their stable match in structured matching markets. In contrast to prior works, the proposed algorithms make decisions based solely on an agent's own history of play and requires no foreknowledge of the firms' preferences. Our algorithms are constructed by splitting up the statistical problem of learning one's preferences, from noisy observations, from the problem of competing for firms. We show that under realistic structural assumptions on the underlying preferences of the agents and firms, the proposed algorithms incur a regret which grows at most logarithmically in the time horizon. However, we note that in the worst case, it may grow exponentially in the size of the market.

## 1 Introduction

Online decision-making under uncertainty is one of the central problems in modern machine learning, reflecting the need for efficient and high performing algorithms for real-time learning in real-world settings. Despite being such a well-researched area, there is a broad lack of understanding of how to deploy online learning algorithms into settings in which they must compete with each other for resources or information. Indeed, while classic problems of online learning deal with trading off the exploration of possible choices and the exploitation of current knowledge (i.e., the exploration-exploitation tradeoff Lattimore and Szepesvári (2020); Slivkins (2019)), the addition of competition adds a new axis upon which algorithms must operate Mansour et al. (2017); Aridor et al. (2020)—namely that of competing (perhaps unsuccessfully) for highly desired outcomes or settling for less desired (but also less competitive) outcomes. Broadly, speaking, the dominant approach to dealing with competition in machine learning remains to treat opponents as adversarialCesa-Bianchi and Lugosi (2006), despite a long literature in economics and game theory Littman (1994); Fudenberg et al. (1998) showing how agents who understand the competitive structure of problems can sometimes vastly outperform solutions based upon worst-case modeling.

In this paper, we address the problem of online learning in competitive settings in the context of *two-sided matching markets*. Two-sided matching markets *match* users on one side of the market

to those on the other to facilitate the exchange of goods or services. In such settings, each user on one side of the market has an inherent preference ordering for the users on the other side of the market. Since each user seeks to find their most desired match, this results in a game in which a natural equilibrium is that of a *stable matching* wherein no two users would prefer switching from their current match to each other given their preferences. In seminal work, Gale and Shapley (1962) proposed a simple and effective algorithm— the *Deferred Acceptance (DA) Algorithm*— that users on one side of the market can implement to find such a solution when every user knows their own preferences. The algorithm has been widely used in examples ranging from kidney exchanges to medical resident matching where preferences can be assigned or reported to a central authority which does the matching. However, recent years have seen the emergence of a new kinds of *online* matching markets like online labor markets (e.g. TaskRabbit, Upwork), online dating markets (e.g. Tinder, Match.com), online crowdsourcing platforms (e.g. Amazon mechanical turk) where the users do not know their preferences apriori, and can repeatedly interact with the market to improve their match quality.

Motivated by these applications we consider a generalization of the problem studied in Gale and Shapley (1962) wherein one side of the market— the agents— do not know their own preferences, but are able to interact repeatedly with the market. In particular, we analyze a repeated game in which, at each round, agents can request to match with a user or firm on the other side of the market. If, at a given round, multiple agents request the same firm, the firm— assumed to be a myopic utility maximizer— accepts the request of its most preferred agent (who receives a noisy measurement of their utility of the match from which they can learn their preferences) and rejects the others (who receive no information about their preferences). This setup has been studied in a line of recent works on online matching markets Liu et al. (2020, 2021); Sankararaman et al. (2021); Basu et al. (2021).

Successful algorithms for this framework must simultaneously solve a statistical learning problem (that of learning about their own preferences) and a competitive problem (ensuring that agents get their most desired match despite the presence of other self-interested agents in the market). Previous works for addressing this problem propose algorithms that are centralized Liu et al. (2020) (whereby agents send their current beliefs over their preferences to a central platform which does the matching), require coordination between agents (i.e., a choreographed set of strategies to minimize rejections) Sankararaman et al. (2021); Basu et al. (2021), or require agents to fully observe the market outcomes of other agents Liu et al. (2021). In contrast, the DA algorithm— which we take to be the full-information benchmark to which we compare algorithms— is (i) fully decentralized, (ii) coordination-free, and (iii) requires agents to make decisions only based upon their own history of rejections and successful matchings. Designing learning algorithms that operate under conditions (i)-(iii) ensures scalability and privacy in large-scale systems where it is unrealistic to assume that agents can keep track of all other agents' matchings. Thus in this work we focus on the question: **Does there exist decentralized and coordination-free algorithms that are based only on local history of interactions and provably converges to stable matching?**

**Contributions.** In this work we design algorithms for learning while matching in a class of structured matching markets known as $\alpha-$reducible matching markets[1]. This condition ensures that there exists an unique stable matching and encompasses many realistic preference structures including serial dictatorship and no crossing conditions Clark (2006). We show that the proposed algorithms incur a stable regret with respect to the unique stable matching that grows at most logarithmically in the time horizon. However, in the worst case it can grow exponentially in the number of agents and firms, which we believe is an artifact of the proof technique. The particular contributions of this paper are:

1. We present a general framework for the construction of decentralized, communication, and coordination-free algorithms for learning while matching. In particular, we combine a index-based stochastic bandit module (in particular the Upper Confidence Bounds algorithm and Thompson Sampling) Auer (2002); Lattimore and Szepesvári (2020); Slivkins (2019) for solving the statistical problem of learning an agent's preferences with a path-length adversarial bandit module Bubeck et al. (2019); Wei and Luo (2018) for dealing with the competitive problem. The resulting algorithms make are fully decentralized, and communication and coordination-free since they make use of only an agent's history of collisions, matches, and rewards to choose which firm

---

[1]The results in this paper extend under the assumption of Sequential Preference Condition (SPC) on the underlying market (ref. Remark 5).

to request at a given time. Furthermore the algorithms are "any-time" algorithms, in that they do not require knowledge of time horizon and do not require any specific parameters of the bandit instance beyond the sub-gaussian parameter of the noise.

2. We show that when the agents' and firms' preferences satisfy the $\alpha-$reducibility condition and *every* agent uses the algorithm, the regret accumulated by any agent $a$ against the stable match is $O\left(\frac{C_a|\mathcal{A}||\mathcal{F}|log(T)}{\Delta^2}\right)$ where $\mathcal{A}$ is the set of agents, $\mathcal{F}$ is the set of firms, $\Delta$ is the minimum sub-optimality gap of any agent in the market, and $C_a$ is a constant that depends on the $\alpha-$reducible structure of the market. We note that in the worst case this constant may grow exponentially with the size of the market.

**Prior work.** The particular intersection of MABs and two-sided matching markets that we analyze has seen a flurry of recent works Liu et al. (2020, 2021); Basu et al. (2021); Sankararaman et al. (2021). To the best of our knowledge, Das and Kamenica (2005), presented the first numerical study on effectively using MAB algorithms to learn preferences in matching markets. However, it was only recently that Liu et al. (2020) rigorously formulated the bandit learning problem in the matching markets, and generalized the notion of *regret* from the MAB literature to matching markets in terms of *stable regret*— i.e., the expected cumulative utility benchmarked against the expected cumulative reward that would have been received if everyone in the market requested their match in a certain stable match[2]. Moreover, they proposed a *centralized* UCB-based algorithm that facilitates the matching between agents and firms given each agents' current beliefs over their preferences and history of play, while ensuring that $\mathcal{O}(|\mathcal{A}||\mathcal{F}|\log(T))$ regret for a UCB based algorithm, where $A$ is the number of agents, $F$ is the number of Firms, and $T$ is the time horizon of the problem. In follow up work Liu et al. (2021) proposed a *decentralized* bandit learning algorithm that allows each user to take its decision in a decentralized manner and still "converge" to stable matching while incurring $O(\exp(|\mathcal{F}|^4)\log^2(T))$ regret. However, the algorithm requires the knowledge of outcomes at other firms at every round, leaving algorithms that are based solely on agents' own history of play as an open problem. Concurrently, Sankararaman et al. (2021) proposed an algorithm that works in phases and makes use of communication between agents to coordinate agents' actions. Under this information structure the algorithm achieves $\mathcal{O}\left(|\mathcal{F}|^2|\mathcal{A}|^2\log(T)\right)$ regret. Moreover their guarantees require that firms have homogeneous preference over the agents (also referred as *serial dictatorship*). Follow-up work, Basu et al. (2021) improved the regret for serial dictatorship to $\mathcal{O}\left(|\mathcal{F}||\mathcal{A}|\log(T)\right)$ by proposing a new algorithm. Additionally, they also showed that if the assumption of serial dictatorship is relaxed to a weaker structural condition then they obtain $O(poly(|\mathcal{A}|,|\mathcal{F}|)\log(T))$ regret. Even thought the proposed algorithm in Basu et al. (2021) has decentralization it is a phase based algorithm where the agents act according to a coordinated protocol at some rounds. In this paper we propose a simple, decentralized, communication and coordination free algorithm in which agents make use of their own local information to learn while matching. Unlike previous works Liu et al. (2020, 2021); Sankararaman et al. (2021); Basu et al. (2021) where the algorithms are constructed using a UCB subroutine, we also show that our algorithmic design paradigm can be also based on Thompson Sampling.

**Organization** The paper is organized as follows: In Section 2 we introduce the general problem setup, introduce matching markets and discuss the structural assumptions on the preferences of agents and firms. In Section 3 we present the algorithmic design paradigm along with a specific algorithm. In Section 4 we present the main result about the regret bound along with a brief sketch of the proof. We conclude the paper in Section 5 and also provide some future research directions. Due to space limitations a detailed comparison to related works and the proofs of our results are relegated to the Supplementary material.

## 2 Setting

We define a two-sided market $\mathcal{M}$ as collection of agents $\mathcal{A}$ and firms $\mathcal{F}$. In the setting under consideration, we assume that every agent $a \in \mathcal{A}$ has *unknown* preferences over firms $f \in \mathcal{F}$ which are captured by utilities $u_a(f) \in \mathbb{R}_+$. Moreover, no two firms give the same utility to a given agent,

---

[2]Note that the stable matching need not be unique in general. Thus the stable regret has to be always specified with respect to which stable matching is being used. Typically, in literature two main stable matchings are considered namely *agent optimal stable matching* and *firm optimal stable matching*.

i.e. $u_a(f) \neq u_a(f')$ if $f \neq f'$. We assume that every agent seeks to be matched to their most preferred firm, and that firms have preferences over all the agents which are also captured by utilities $u_f(a)$ for each $a$ and $f$ such that no two agents give same utility to firms i.e. $u_f(a) \neq u_f(a')$. Importantly, we assume that firms know their own preference orderings over agents and that there are more firms than agents, i.e. $|\mathcal{A}| \leq |\mathcal{F}|$. The interaction between agents and firms happens as follows: In each time step $t = 1, \ldots, T$ every agent $a \in \mathcal{A}$ independently *requests* a firm $f_a(t) \in \mathcal{F}$. As the agents request independently, it is possible that more than one agent requests the same firm $f$. For $f \in \mathcal{F}$, let $\mathbb{A}_f(t) := \{a \in \mathcal{A} : f_a(t) = f\}$ denote the set of agents that request firm $f$ at time step $t$. At each time step $t$, we assume that the firm $f$ accepts the request of their *most preferred* agent in $\mathbb{A}_f(t)$ denoted by $a_f(t) := \arg\max_{a \in \mathbb{A}_f(t)} u_f(a)$, and *rejects* the request of all other agents. That agent $a_f(t)$ is said to be the agent who got *matched* with firm $f$ at time $t$. Moreover every matched agent receives a noisy measurement of their utility, denoted $U_{\mathbf{a},f}$ such that $U_{\mathbf{a},f} = u_{\mathbf{a}}(f) + \zeta_{\mathbf{a},f}$, where $\zeta_{\mathbf{a},f}$ is a zero-mean, one-sub-Gaussian random variable[3]. Meanwhile, all the agents that are rejected are said to have *collided* on firm $f$, for which they receive no utility i.e. $U_{a,f}(t) = 0$.

We restrict that agents *only* receive the following information at any time step $t$:

1. $Y_a(t) = \mathbb{1}(a \text{ is matched to } f_a(t))$, which captures if agent $a$ gets matched at time $t$

2. if they get matched, the noisy measurement of their utility, $U_{a,f}(t)$.

**Remark 1.** *We note that in this setup an agent does not know* anything *about how other agents are performing in the market. Agents do* not *observe any information about the matching and collisions of other agents. We remark that this is* the same *information structure as that assumed by the DA algorithm and is the key assumption that differentiates our work from prior work on this problem Liu et al. (2020, 2021); Basu et al. (2021); Sankararaman et al. (2021).*

In the following subsection, we recall some important results from matching market literature that are crucial to further exposition.

## 2.1 Preliminaries on matching markets

To analyze the matching market defined in the previous section we recall key concepts from the literature on matching markets. A matching $\mathbb{M} : \mathcal{A} \longrightarrow \mathcal{F}$ is an injective function such that $\mathbb{M}(a) = f$ denotes that $a$ and firm $f$ are matched. We call a matching *unstable* if there is an agent-firm tuple $(a, f) \in \mathcal{A} \times \mathcal{F}$ such that $u_a(\mathbb{M}(a)) < u_a(f)$ and $u_f(a) > u_f(\mathbb{M}^{-1}(f))$. In words, there is a pair $(a, f)$ who both prefer each other over their current match, such pair is called a *blocking pair*. A matching is *stable* if it is not unstable. It is usually the case that a market may have multiple stable matchings. However, for the purpose of this paper we focus on markets which are $\alpha-$reducible, first introduced in Alcalde (1994) and further analyzed in Clark (2006), that ensures there is a unique stable matching. Before formally describing this property we introduce the notion of a submarket and fixed pair.

A sub-market of $\mathcal{M}$ is a market $\mathcal{M}'$ such that $\mathcal{M}' = \mathcal{A}' \cup \mathcal{F}'$ where $\mathcal{A}' \subseteq \mathcal{A}$, $\mathcal{F}' \subseteq \mathcal{F}$, and $|\mathcal{A}'| \leq |\mathcal{F}'|$. Meanwhile, a pair $(a, f) \in \mathcal{A} \times \mathcal{F}$ is a *fixed pair* of market $\mathcal{M}$ if $u_a(f) \geq u_a(f')$ for all $f' \in \mathcal{F}$ and $u_f(a) \geq u_f(a')$ for all $a' \in \mathcal{A}$. In words, a fixed pair is any agent-firm pair that prefer each other over any other options in the market. We now define the notion of $\alpha-$reducibility.

**Definition 2** ($\alpha$-reducibility). *A market $\mathcal{M} = \mathcal{A} \cup \mathcal{F}$ is $\alpha$-reducible if every sub-market of $\mathcal{M}$ has a fixed pair.*

One important class of markets which satisfy $\alpha-$reducibility is that where one side of the market has uniform preferences over the other side. We note that every sub-market of of $\mathcal{M}$ has a unique stable matching if $\mathcal{M}$ is $\alpha$-reducible Clark (2006). The preceding property of $\alpha-$reducible markets will be crucial to obtain regret guarantees for the proposed algorithm in this paper. Thus, we assume that $\mathcal{M}$ is $\alpha$-reducible[4].

**Remark 3.** *An important property of $\alpha-$reducibility assumption that is central to the subsequent analysis is that it allows us to partition the market into various sub-markets by sequentially eliminating*

---

[3]Here we assume that the random noise is appropriately bounded such that $U_{\mathbf{a},f} \geq 0$ for all $(a, f)$

[4]We can also handle the scenarios where the underlying preferences satisfies SPC conditionClark (2006). More discussion is provided in Remark 5.

*fixed pairs. More formally, lets define $\mathcal{A}_0 = \mathcal{F}_0 = \varnothing$ and $\mathcal{M}_0 = \mathcal{M}$. Now for $i \geq 1$ lets define inductively $\mathcal{A}_i \subseteq \mathcal{A} \setminus \{\cup_{j=1}^i \mathcal{A}_{j-1}\}, \mathcal{F}_i \subseteq \mathcal{F} \setminus \{\cup_{j=1}^i \mathcal{F}_{j-1}\}$ be the set of agents and set of firms that constitute fixed pair in market $\mathcal{M}_{i-1}$. That is, for every agent $a \in \mathcal{A}_i$ there exists a unique $f \in \mathcal{F}_i$ such that $(a, f)$ is a fixed pair of market $\mathcal{M}_{i-1}$. The iteration evolves as $\mathcal{M}_i :=$ $\{\mathcal{A} \setminus \{\cup_{j=0}^i \mathcal{A}_j\}\} \cup \{\mathcal{F} \setminus \{\cup_{j=0}^i \mathcal{F}_j\}\}$. Let $K$ be the total number of such sub-markets $\{\mathcal{M}_i\}$. Moreover such decomposition of market is unique.*

For any agent $a \in \mathcal{A}$ we denote by $f_a^*$ its match in the unique stable matching. Furthermore, let $\bar{\mathbb{F}}_a := \{f \in \mathcal{F} : u_a(f) > u_a(f_a^*)\}$ be the set of firms that agent $a$ prefers over its stable match. We call such firms *super-optimal* firms for $a$. Similarly, let $\underline{\mathbb{F}}_a := \{f \in \mathcal{F} : u_a(f) < u_a(f_a^*)\}$ be the set of firms which are less preferred than the stable match by agent $a$. We call such firms *sub-optimal* firms for $a$. Note that we have following lemma which states a crucial property of super-optimal firms for $\alpha-$reducible markets.

**Lemma 4.** *For any $i \in [K]$ and agent $a \in \mathcal{A}_i$ the set of super-optimal firms are contained in $\cup_{j=1}^{i-1} \mathcal{F}_j$.*

An immediate conclusion of Lemma 4 is that it creates a hierarchy in the market. That is, an agent $a \in \mathcal{A}_i$, for some $i \in [K]$, is in a sense "higher ranked" than a agent $a' \in \mathcal{A}_j$ for $j > i$ as the former's stable match can be super-optimal for the latter. This sort of hierarchy naturally manifests itself in the learning process where learning of agent $a$ creates *externality* for agent $a'$.

Before we formally present the algorithm, we make the following remark about the preference structure for which the results in this article hold.

**Remark 5.** *The notion of $\alpha$-reducibility is weaker than the* no crossing condition *and serial dictatorship Clark (2006). These conditions have been introduced in the effort to characterize the existence and uniqueness of a stable matching. We note that $\alpha$-reducibility is a stricter condition than Sequential Preference Condition (SPC) Clark (2006). However, the results presented in this work extend directly under the assumption of SPC. This is because all of the results are derived based on the decomposition stated in Remark 3 which can be also obtained from the definition of SPC Clark (2006); Karpov (2019). However, as pointed out by Clark (2006), given any population $P$ of agents alpha-reducibility is necessary and sufficient condition for unique stable matching regardless of subpopulation sampled from $P$. However, this property does not hold for SPC.*

For ease of reference, all key notations used in paper are presented in a table in the Supplementary material.

## 3 Algorithms

In this section we present a novel algorithm design principle for agents to learn about the preferences while ensuring that they perform competitively against the match that they could have achieved if they knew their preferences and used the DA algorithm. Throughout this section, we assume that every agent $a \in \mathcal{A}$ uses these algorithms in order to decide which firm to choose at any time $t$. The proposed algorithms—by design— make use of only the feedback information outlined in (1)-(2) in Section 1, and have no implicit or explicit communication and coordination strategies like e.g., phase based strategies with coordinated actions Basu et al. (2021) or partial observation of actions of other agents Liu et al. (2021) etc. Thus, the algorithms operate in the same regime as the DA algorithm, but without the assumption that agents know their preferences. Key to our approach, is the blending stochastic bandit (SB) algorithms with an adversarial bandit (AB) algorithms. In the subsequent exposition we will formally describe our approach and show its desirable properties in terms of regret and convergence.

Before doing so, however, we comment on the difficulties of the problem at hand, and what makes the analysis of these algorithms highly non-trivial. The key challenge in designing algorithms for matching while learning is understanding when to stop requesting *super-optimal* firms (i.e. firms that they prefer more than their stable match) without any foreknowledge of the market structure. The crux of this problem is having an agent learn that certain firms are unattainable due to competition despite the non-stationarity in the environment stemming from fact that other agents are learning simultaneously and not knowing who they collide with and who is successfully getting matched at each round. Furthermore, due to a lack of communication or coordination, agents cannot learn about which firms are super-optimal without risking many collisions.

A sketch of the algorithm is described in words in Algorithm 1, and the exact algorithm for the setting in which agents use the UCB algorithm as a subroutine is presented in Algorithm 2. A corresponding version of algorithm with Thompson sampling based stochastic subroutine is presented in Supplemental material. As per Algorithm 2, each agent is equipped with a stochastic bandit (SB)

---

**Algorithm 1:** High-level algorithmic description

Each agent $a \in \mathcal{A}$ at every time $t \in [T]$:

1. Keeps an ordering of firms as per an index-based stochastic bandit subroutine

2. Agent $a$ goes over the firms as per the ordering one by one

3. Using an adversarial bandit subroutine decides whether to *request* the firm or to *prune it*

    (a) If a firm is requested then agent either gets matched or gets collided

    (b) If pruned then then the agent moves to next firm as per the ordering in Step 1.

4. Updates the stochastic and adversarial bandit subroutine based on the feedback received

---

subroutine. At every time step $t \in [T]$, the SB subroutine of every agent $a$ maintains ordering of firms in decreasing order of preferences according to an index (e.g. UCB). We denote this index of firm $f$ as maintained by agent $a$ as $\mathsf{UCB}_{a,f}(t)$. Next, at that time step, every agent *considers* each firm one by one in decreasing order of $\mathsf{UCB}_{a,f}(t)$. For any firm $f$ considered by agent $a$ at time $t$, the agent makes a decision to either *request* $f$ or to *prune*[5] it (that is, to reject that firm). In particular, agent $a$ requests firm $f$ with probability $p_{a,f}(t)$. Let $P_{a,f}(t) \sim \text{Bernoulli}(p_{a,f}(t))$. If a firm is pruned (i.e. $P_{a,f}(t) = 0$) then the next best firm from the sorted list is chosen and the process continues until a firm is requested (i.e. $P_{a,f}(t) = 1$). However, if all of the firms are pruned then at that time instant the agent simply requests the firm $\arg\max_f \mathsf{UCB}_{a,f}(t)$. Once an agent decides which firm to request, it obtains a noisy utility if it gets successfully matched. This feedback is used by the agent to update its UCB-index. Based on whether an agent $a$ decides to prune or request a particular firm $f$, it updates $p_{a,f}$ using an AB subroutine. The details about this are stated in Section[6] 3.2 We note that all firms are not considered by agent $a$ at every time $t$. Once an agent decides to request a firm $f$, it does not consider firms in the set $\{f' \in \mathcal{F} : \mathsf{UCB}_{a,f'}(t) < \mathsf{UCB}_{a,f}(t)\}$. Formally, for any agent-firm tuple $(a, f) \in \mathcal{A} \times \mathcal{F}$ let the event that the agent $a$ considers the firm $f$ at time $t$, to decide whether to request it or prune it, be denoted by $E_{a,f}^{(\mathsf{c})}(t) = \mathbb{1}\,(P_{a,f'}(t) = 0, \quad \forall\, f' : \mathsf{UCB}_{a,f}(t) \leq \mathsf{UCB}_{a,f'}(t))$. If a firm $f$ is considered by agent $a$ then the event when agent $a$ requests $f$ is denoted by $E_{a,f}^{(\mathsf{r})}(t) = \mathbb{1}\left(P_{a,f}(t) = 1, E_{a,f}^{(\mathsf{c})}(t) = 1\right)$.

In Section 3.1 we describe the UCB computation method for the SB subroutine. Finally, in Section 3.2, we illustrate how the matchings and collisions are used to update the probability $p_{a,f}(t)$ as per an AB subroutine.

### 3.1 Stochastic Bandit Subroutine

The stochastic bandit subroutine is used to efficiently deal with inherent uncertainty in the payoff obtained upon successful matching. In this section we develop the theory for the setting in which agents use a UCB based SB subroutine. Similar results for Thompson Sampling are supplied in the supplementary material.

To begin, we denote the number of times agent $a$ gets successfully matched with firm $f$ till time $t$ as $M_{a,f}(t)$. Similarly, the number of times agent $a$ gets collided with firm $f$ till time $t$ be $C_{a,f}(t)$. Given this notation, the UCB Auer (2002) estimate of agent $a$ for every $f$ at time $t$ is given by

$$\mathsf{UCB}_{a,f}(t) = \hat{\mu}_{a,f}(t-1) + \sqrt{\frac{2\log(1 + \bar{M}_a \log^2(\bar{M}_a))}{M_{a,f}(t)}},$$

---

[5]Note that by pruning here we do not mean permanent pruning, it is used to describe that a particular firm is not consider at that time step

[6]The corresponding algorithmic subroutine $\mathsf{AB\_Subroutine}$ is presented in the Supplementary material.

**Algorithm 2:** UCB based Decentralized Matching Algorithm (UCB-DMA)

---

**Initialize:** $\hat{\mu}_{a,f} = 0, M_{a,f} = 0, p_{a,f} = 0.5, x_{a,f} = 0.5, L_{a,f} = 0, \ \forall a \in \mathcal{A}, f \in \mathcal{F}$

1   **for** $t = 1, \ldots, T$ **do**
2     **for** $f \in \mathcal{F}$ **do**
3       Set $\mathsf{UCB}_{a,f} = \hat{\mu}_{a,f} + \sqrt{\frac{2\log(1+(\bar{M}_a)\log^2(\bar{M}_a))}{M_{a,f}}}$, where $\bar{M}_a = \sum_{f \in \mathcal{F}} M_{a,f}$
4     **end**
5     Set $\mathsf{ArgUCB}_a = \mathsf{ArgDescendingSort}(\{\mathsf{UCB}_{a,f}\}_{f \in \mathcal{F}})$ and $i = 1$
6     **while** $i \leq |\mathcal{F}|$ **do**
7       Set $f = \mathsf{ArgUCB}_a^{[i]}$ and sample $P_{a,f} \sim \mathsf{Bernoulli}(p_{a,f})$
8       **if** $P_{a,f} = 0$ **then**
9         Update $(x_{a,f}, p_{a,f}, L_{a,f}) \longleftarrow \mathsf{AB\_Subroutine}(P_{a,f}, x_{a,f}, p_{a,f}, L_{a,f}, Y_a)$
10       **end**
11       **if** $P_{a,f} = 1$ **then**
12         Request firm $f$ and receive $(U_a, Y_a)$
13         Update $\hat{\mu}_{a,f} \longleftarrow Y_a \frac{\hat{\mu}_{a,f} M_{a,f} + U_a}{M_{a,f}+1} + (1-Y_a)\hat{\mu}_{a,f}, \ \ M_{a,f} \longleftarrow M_{a,f} + Y_a,$
14         Update $(x_{a,f}, p_{a,f}, L_{a,f}) \longleftarrow \mathsf{AB\_Subroutine}(P_{a,f}, x_{a,f}, p_{a,f}, L_{a,f}, Y_a)$
15         break while;
16       **end**
17       $i \longleftarrow i + 1$
18     **end**
19     **if** $i = |\mathcal{F}| + 1$ **then**
20       Request firm $\mathsf{ArgUCB}_a^{[1]}$ and receive $(U_a, Y_a)$
21       Update $\hat{\mu}_{a,f} \longleftarrow Y_a \frac{\hat{\mu}_{a,f} M_{a,f} + U_a}{M_{a,f}+1} + (1-Y_a)\hat{\mu}_{a,f}, \ \ M_{a,f} \longleftarrow M_{a,f} + Y_a$
22     **end**
23 **end**

---

where $\bar{M}_a(t) = \sum_{f \in \mathcal{F}} M_{a,f}(t)$ and $\hat{\mu}_{a,f}(t-1)$, are the total number of successful matches for agent $a$ and the empirical average of the payoffs received from successfully matching to firm $f$ until time $t$ respectively. The UCB estimate is composed of two parts: (i) the empirical mean which captures the exploitation aspect; and (ii) exploration bonus that decreases as $M_{a,f}(t)$ increases. We remark that it does not depend on the number of collisions $C_{a,f}(t)$.

### 3.2   Adversarial Bandit Subroutine

A key component of the proposed methodology is to use an adversarial bandit subroutine to deal with the competitive aspect of the problem. In particular, the AB subroutine updates the request probability $(p_{a,f})_{f \in \mathcal{F}}$ such that agent stops requesting firm on which the collisions are high (but ensures that it does not miss out on the firm if it is achievable). Intuitively, by construction, the adversarial bandit algorithm learns to prune firms on which collisions would happen frequently, and request firms where it is possible to successfully match very often. We show this by analyzing its regret and showing that high regret is incurred if the algorithm either prunes too often when successfully matching is possible or requesting a firm that is unachievable due to the frequent presence of higher ranked agents. By bounding the regret of the AB subroutine we immediately get a bound on the number of collisions.

We now describe the update scheme for $p_{a,f}(t)$ for any $(a, f)$ at any time $t \in [T]$. In this work we consider an optimistic mirror descent based AB subroutine specialized from Bubeck et al. (2019). Interestingly such AB algorithms have data dependent regret bounds Wei and Luo (2018); Bubeck et al. (2019) unlike other AB algorithms like Exp3 Lattimore and Szepesvári (2020); Slivkins (2019). Since the competition in the matching market is not actually adversarial such data-dependent regret bounds enables us characterize the competition more effectively in the analysis than just treating

competition as adversarial[7]. We note that the proof techniques developed here can also be used to analyze an Exp3 based AB subroutine but the regret bounds of such an approach will not be as sharp.

For a given agent $a$, our algorithm associates a separate AB subroutine to every firm $f \in \mathcal{F}$. Each AB algorithm has *two arms* which correspond to the action of requesting the firm $f$ or pruning it, each of which incurs different losses depending. In particular, if $P_{a,f}(t) = 0$ then it receives a fixed loss of 0; if $P_{a,f}(t) = 1$ the loss received is $+1$ or $-1$ if it collides or matches respectively. If we denote the loss received by the AB subroutine associated with $(a, f)$ at time $t$ by $L_{a,f}(t)$, we note that $L_{a,f}(t) = P_{a,f}(t)(1 - 2Y_a(t))$. Note that $Y_a(t)$ is unknown to any agent before requesting any firm as it also depends on the requests made by other agents.

We note that the request probability $p_{a,f}$ is not updated at every time $t$, but only when $E^{(c)}_{a,f}(t) = 1$ (i.e., if all firms with a higher UCB index have been pruned). As such the adversarial bandit algorithms can be seen as operating on a random timescale $\tau_{a,f}(T) = \{t \in [T] : E^{(c)}_{a,f}(t) = 1\}$ which are the time steps on which agent $a$ considers firm $f$. We note that for any $t \notin \tau_{a,f}(T)$ we have $p_{a,f}(t+1) = p_{a,f}(t)$.

For the specific AB algorithm we analyze (which is a version of optimistic mirror descent with a log-barrier regularizer first analyzed in Wei and Luo (2018)), the simple setup of the losses leads to a closed form update for the probability of requesting or pruning a firm. In particular, for every $(a, f) \in \mathcal{A} \times \mathcal{F}$ and $t \in \tau_{a,f}(T)$, the optimistic mirror descent AB subroutine creates an unbiased estimate of the loss due to pruning and requesting as $\hat{L}^{(\text{prune})}_{a,f}(t)$ and $\hat{L}^{(\text{pull})}_{a,f}(t)$ respectively. In particular, if $P_{a,f}(t) = 1$

$$\hat{L}^{(\text{prune})}_{a,f}(t) = \frac{1 + L_{a,f}(t-1)}{2}, \quad \hat{L}^{(\text{pull})}_{a,f}(t) = \frac{1 - 2Y_a(t) - L_{a,f}(t-1)}{2p_{a,f}(t)} + \frac{1 + L_{a,f}(t-1)}{2}.$$

On the other hand, if $P_{a,f}(t) = 0$ then

$$\hat{L}^{(\text{prune})}_{a,f}(t) = \frac{-L_{a,f}(t-1)}{2(1 - p_{a,f}(t))} + \frac{1 + L_{a,f}(t-1)}{2}, \quad \hat{L}^{(\text{pull})}_{a,f}(t) = \frac{1 + L_{a,f}(t-1)}{2}$$

The term $\frac{1 + L_{a,f}(t-1)}{2}$ is an optimistic prediction of the losses based on the last round of interaction Bubeck et al. (2019). Given these estimators the probability of requesting a firm is updated as:

$$p_{a,f}(t+1) = (1 - \Lambda_{a,f}(t))x_{a,f}(t) + \Lambda_{a,f}(t)P_{a,f}(t),$$

where:

$$x_{a,f}(t) = \left(2 + \xi(t) - \sqrt{4 + \xi(t)^2}\right)(2\xi(t))^{-1}$$

for $\xi(t) = \eta\left(\hat{L}^{(\text{pull})}_{a,f}(t) - \hat{L}^{(\text{prune})}_{a,f}(t)\right) + \frac{1}{x_{a,f}(t-1)} - \frac{1}{1 - x_{a,f}(t-1)}$, is the result of a step of mirror descent with the log-barrier regularizer, and $\Lambda_{a,f}(t) = \frac{\lambda(1 - L_{a,f}(t))}{2 + \lambda(1 - L_{a,f}(t))}$, for $\lambda > 0$, promotes exploration. The algorithmic description of this process is stated in the Supplementary material.

## 4 Regret bounds for decentralized matching algorithms

To capture the performance of the algorithm we use the natural notion of *stable regret* as introduced in Liu et al. (2020). More formally, the stable regret accrued by any agent $a \in \mathcal{A}$ is

$$\mathbb{E}[\mathcal{R}_a(T)] = \mathbb{E}\left[\sum_{t=1}^{T} u_{a,f^*_a} - \sum_{t=1}^{T} u_{a,f_a(t)}\right] \leq \sum_{f \in \mathbb{F}_a} \Delta_a(f)\mathbb{E}[M_{a,f}(T)] + u_a(f^*_a)\sum_{f \in \mathcal{F}} \mathbb{E}[C_{a,f}(T)],$$

(4.1)

where $\Delta_a(f) = u_a(f^*_a) - u_a(f)$ is the gap between the mean that agent $a$ gets upon successfully matching with its stable match as compared firm $f$. If there are no collisions, then this regret definition is same as that used in stochastic bandits literature (Lattimore and Szepesvári (2020)). In the following theorem, we present the regret of any agent using Algorithm 2:

---

[7]We review the required background on optimistic mirror descent based AB algorithms in the Supplementary material along with a result which captures the corresponding data-dependent regret bounds in the setting of matching markets.

**Theorem 6.** *Suppose every agent $a \in \mathcal{A}$ uses Algorithm 2. Then for any $i \in [K]$ :*

$$\sum_{j=1}^{i} \sum_{a \in \mathcal{A}_j} \mathbb{E}[\mathcal{R}_a(T)] = \mathcal{O}\left(C_i |\mathcal{F}||\mathcal{A}| \log(T) \left(1 + \frac{1}{\Delta^2}\right)\right)$$

*where $\Delta = \min_{a,f} \Delta_{a,f}$ and $C_i = i\theta^i$ for some positive scalar $\theta > 1$. Note that $C_1 < C_2 < ... < C_K$.*

We see that the regret of any agent $a \in \mathcal{A}$ is logarithmic in horizon $T$, which matches the lower bound for single player stochastic bandit algorithms Lai and Robbins (1985). As such, perhaps surprisingly, we observe that in $\alpha$-reducible markets, it is possible for agents to learn while competing without incurring drastically worse regret in the long run. It is interesting to note that the learning of agent depends on its position in the market as per preferences (Remark 3). An agent low in the hierarchy incurs more regret during the learning process due to the agents higher up in the hierarchy driven mainly by the larger number of collisions incurred while waiting for agents higher in the hierarchy to stop exploring. We note that in the worst case the constant $C_i$ can grow exponentially in the number of agents in the market. We note that this is a consequence of the proof technique and not fundamental limitation of the algorithmic design paradigm as we show in the supplementary material through numerical studies. We leave this as a future work to establish tighter regret bounds in terms of number of agents. In the supplementary material we also show that in Algorithm 2 if we use a SB subroutine based on Thompson Sampling then a similar regret guarantee can be obtained. We now present a sketch of the proof of Theorem 6.

**Sketch of the proof.** Before presenting the sketch, we first define few notations that would make the exposition clear. Let $M_{a,\underline{\mathbb{F}}_a}(T) = \sum_{f \in \underline{\mathbb{F}}_a} M_{a,f}(T)$, $M_{a,\bar{\mathbb{F}}_a}(T) = \sum_{f \in \bar{\mathbb{F}}_a} M_{a,f}(T)$. Moreover, for any $a \in \mathcal{A}$ define $H_{a,f_a^*}(t) = \{\exists a' \in \mathcal{A} \text{ s.t. } u_{f_a^*}(a') \geq u_{f_a^*}(a), f_{a'}(t) = f\}$ which is an event that characterizes if any other more preferred agent has requested the stable match of agent $a$ at time $t$. Against the preceding backdrop, we now present the following crucial lemma:

**Lemma 7.** *Suppose every agent uses Algorithm 2*

*(L1) For any $i \in [K]$, the cumulative regret can be decomposed as*

$$\sum_{j=1}^{i} \sum_{a \in \mathcal{A}_j} \mathbb{E}[\mathcal{R}_a(T)] = \mathcal{O}\left(\sum_{i=1}^{k} \sum_{a \in \mathcal{A}_i} (\mathbb{E}[M_{a,\underline{\mathbb{F}}_a}(T)] + \sum_{\substack{f \in F \\ f \neq \{f_a^*\}}} \mathbb{E}[C_{a,f}(T)] + \mathbb{E}[\sum_{t=1}^{T} H_{a,f_a^*}(t)])\right);$$

*(L2) For any $i \in [K]$, the expected matches with suboptimal firm satisfies*

$$\sum_{j=1}^{i} \sum_{a \in \mathcal{A}_j} \mathbb{E}[M_{a,\underline{\mathbb{F}}_a}(T)] = \mathcal{O}\left(\sum_{j=1}^{i} \sum_{a \in \mathcal{A}_j} \left(|\underline{\mathbb{F}}_a| \log(T)\left(1 + \frac{1}{\Delta^2}\right) + \mathbb{E}\left[\sum_{t=1}^{T} H_{a,f_a^*}(t)\right]\right)\right)$$

*(L3) The expected number of collisions between for any agent $a \in \mathcal{A}$ satisfies*

$$\sum_{f \in \mathcal{F}} \mathbb{E}[C_{a,f}(T)] = \mathcal{O}\left(|\mathcal{F}| \log(T) + \mathbb{E}\left[M_{a,\underline{\mathbb{F}}_a}(T) + M_{a,\bar{\mathbb{F}}_a}(T) + \sum_{t=1}^{T} \mathbb{1}\left(H_{a,f_a^*}(t)\right)\right]\right);$$

*(L4) For any $i \in [K]$ we have*

$$\sum_{j=1}^{i} \sum_{a \in \mathcal{A}_j} \mathbb{E}\left[\sum_{t=1}^{T} \mathbb{1}\left(H_{a,f_a^*}(t)\right)\right] = \mathcal{O}\left(C_i \left(\sum_{j=1}^{i} |\mathcal{A}_j|\right) \log(T)\left(1 + \frac{1}{\Delta^2}\right)\right),$$

*where $C_i$ is a constant dependent on market $\mathcal{M}_i$ such that $C_1 < C_2 < ... < C_K$.*

*(L5) For any $i \in [K]$ we have*

$$\sum_{j=1}^{i} \sum_{a \in \mathcal{A}_j} \sum_{f \in \bar{\mathbb{F}}_a} \mathbb{E}[M_{a,f}(T)] \leq \mathcal{O}\left(C_i \left(\sum_{j=1}^{i} |\mathcal{A}_j|\right) |\mathcal{F}| \log(T)\left(1 + \frac{1}{\Delta^2}\right)\right)$$

Theorem 6 is proved using **(L1)**-**(L5)** from Lemma 7. Note that **(L1)** follows from (4.1) and the definition of $H_{a,f_a^*}(t)$. From **(L1)** we see that to bound the regret we need to consider three components: (i) expected number of matchings with suboptimal firms, (ii) expected number of collisions with any firm other than stable match, (iii) the *potential collisions* at the stable match[8]. **(L2)** bounds the expected number of matchings with suboptimal firms. Note that the total matchings between agent $a$ and firm $f$ is $M_{a,f}(T) = \sum_{t=1}^{T} \mathbb{1}(Y_a(t) = 1, f_a(t) = f)$. Thus, we present the following lemma which plays a key role in the proof of **(L2)**:

**Lemma 8.** *The event that agent $a$ chooses the firm $f \in \underline{\mathbb{F}}_a$ and successfully matches at time $t \in [T]$ satisfies*

$$\{Y_a(t) = 1, f_a(t) = f\} \subset \{Y_a(t) = 1, \textbf{UCB}_{a,f_a^*}(t) \leq \textbf{UCB}_{a,f}(t)\} \cup \{E_{a,f}^{(r)}(t) = 1, E_{a,f_a^*}^{(r)}(t) = 0\}$$

Lemma 8 separates the challenge associated with uncertainty and that of competition. Note that the first event on the right hand side is the one which is standard to the analysis of UCB algorithm (Lattimore and Szepesvári (2020)). Meanwhile, the other event corresponds to the case when the stable firm is pruned by agent $a$ in order to avoid potential collisions. To bound latter event we use the regret bounds for the adversarial bandit subroutine (refer to Appendix).

To bound **(L3)** we use the path length based regret bounds Bubeck et al. (2019); Wei and Luo (2018) for the adversarial bandit subroutine. Meanwhile to bound **(L4)** we use the $\alpha-$reduciblity assumption and **(L2)**. In particular, the $\alpha-$reduciblity assumption induces a hierarchy in the market as per Remark 3. This decomposition reduces the bound in **(L4)** to appropriate accounting of number of matches with suboptimal firms via an induction argument. Finally, **(L5)** follows again due to hierarchy induced by $\alpha-$reducibility and using **(L2)**-**(L4)**.

# 5 Conclusions

We consider a problem of bandit learning in two-sided matching markets comprising of agents and firms. We consider the setting where agents have unknown preferences over the firms. In this paper, we present simple design principle for decentralized, communication and coordination free algorithm for learning in two-sided matching markets. The primary challenge in learning in two-sided matching market is to balance exploration, exploitation and collision avoidance. We embed the aforementioned properties in the algorithm by a novel idea of blending a stochastic bandit subroutine with an adversarial bandit subroutine. The stochastic bandit subroutine is required for balancing the exploration-exploitation trade-off while the adversarial bandit subroutine limits the collisions. As an instance of this design principle, we present an algorithm which has the stochastic bandit subroutine based on UCB and the adversarial bandit subroutine based on Optimistic Mirror Descent algorithm. We show that if the preferences of agents satisfy certain structure known as $\alpha$-reducibility (or SPC condition), then these algorithms incur a regret which is logarithmic in the time horizon. However, in the worst case, the regret may grow exponentially in the size of the market. We believe that this is an artifact of proof technique and not the limitation of algorithmic design.

There are several directions in which this work can be extended in future. First, it would be an interesting avenue of future research to improve the dependence of regret bound on size of the market. We believe the worst case exponential dependence on number of agents is an artifact of the current proof technique. Second, it would be also interesting to relax the imposed structure on matching markets. Specifically, it would be interesting to extend the results to the setting when the underlying preferences satisfy $\alpha-$condition Karpov (2019) which is a necessary and sufficient condition for uniqueness of stable matching. Lastly, it would be also an interesting direction of research to develop decentralized algorithms when both sides of the market are simultaneously learning. The scenario where firms are also learning their preferences in non-trivial extension of this work as the firms may incorrectly prefer suboptimal firms which may in turn lead to wrong feedback to agents.

## Acknowledgments and Disclosure of Funding

Research is supported in part by NSF grant DMS 2013985 "THEORINet: Transferable, Hierarchical, Expressive, Optimal, Robust and Interpretable Networks" and C3.ai Digital Transformation Institute.

---

[8]by potential collision at stable match we mean total number of collision that would have been faced by an agent at its stable firm had it always requested the stable firm

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
