\left\{Y_a(t) = 1, \textbf{\textit{UCB}}_{a,f_a^*}(t) \leq \textbf{\textit{UCB}}_{a,f}(t)\right\} \cup \{E_{a,f}^{(r)}(t) = 1, E_{a,f_a^*}^{(r)}(t) = 0\}$$

Lemma 8 separates the challenge associated with uncertainty and that of competition. Note that the first event on the right hand side is the one which is standard to the analysis of UCB algorithm (Lattimore and Szepesvári (2020)). Meanwhile, the other event corresponds to the case when the stable firm is pruned by agent $a$ in order to avoid potential collisions. To bound latter event we use the regret bounds for the adversarial bandit subroutine (refer to Appendix).

To bound **(L3)** we use the path length based regret bounds Bubeck et al. (2019); Wei and Luo (2018) for the adversarial bandit subroutine. Meanwhile to bound **(L4)** we use the $\alpha-$reduciblity assumption and **(L2)**. In particular, the $\alpha-$reduciblity assumption induces a hierarchy in the market as per Remark 3. This decomposition reduces the bound in **(L4)** to appropriate accounting of number of matches with suboptimal firms via an induction argument. Finally, **(L5)** follows again due to hierarchy induced by $\alpha-$reducibility and using **(L2)**-**(L4)**.

## 5    Conclusions

We consider a problem of bandit learning in two-sided matching markets comprising of agents and firms. We consider the setting where agents have unknown preferences over the firms. In this paper, we present simple design principle for decentralized, communication and coordination free algorithm for learning in two-sided matching markets. The primary challenge in learning in two-sided matching market is to balance exploration, exploitation and collision avoidance. We embed the aforementioned properties in the algorithm by a novel idea of blending a stochastic bandit subroutine with an adversarial bandit subroutine. The stochastic bandit subroutine is required for balancing the exploration-exploitation trade-off while the adversarial bandit subroutine limits the collisions. As an instance of this design principle, we present an algorithm which has the stochastic bandit subroutine based on UCB and the adversarial bandit subroutine based on Optimistic Mirror Descent algorithm. We show that if the preferences of agents satisfy certain structure known as $\alpha$-reducibility (or SPC condition), then these algorithms incur a regret which is logarithmic in the time horizon. However, in the worst case, the regret may grow exponentially in the size of the market. We believe that this is an artifact of proof technique and not the limitation of algorithmic design.

There are several directions in which this work can be extended in future. First, it would be an interesting avenue of future research to improve the dependence of regret bound on size of the market. We believe the worst case exponential dependence on number of agents is an artifact of the current proof technique. Second, it would be also interesting to relax the imposed structure on matching markets. Specifically, it would be interesting to extend the results to the setting when the underlying preferences satisfy $\alpha-$condition Karpov (2019) which is a necessary and sufficient condition for uniqueness of stable matching. Lastly, it would be also an interesting direction of research to develop decentralized algorithms when both sides of the market are simultaneously learning. The scenario where firms are also learning their preferences in non-trivial extension of this work as the firms may incorrectly prefer suboptimal firms which may in turn lead to wrong feedback to agents.

## Acknowledgments and Disclosure of Funding

Research is supported in part by NSF grant DMS 2013985 "THEORINet: Transferable, Hierarchical, Expressive, Optimal, Robust and Interpretable Networks" and C3.ai Digital Transformation Institute.

---

[8]by potential collision at stable match we mean total number of collision that would have been faced by an agent at its stable firm had it always requested the stable firm

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

# Supplemental Materials for "Decentralized Communication and Coordination-Free Learning in Matching Markets"

The supplemental material is organized as follows: In Section A, we present an extended review of existing literature. In Section B, we review the relevant results in the area of adaptive adversarial bandits. In Section C, we present the proofs of the main lemmas stated in the main paper. In Section D, we present the proof of the Theorem 6. In Section E, we present some technical lemmas, along with their proofs, which are crucial to the proofs in Section C and Section D. In Section F we present the results when instead of UCB based stochastic module we use Thompson sampling based stochastic module in Algorithm 2 (refer Algorithm 5). In Section G, we present the results of numerical experiments conducted using Algorithm 2 and Algorithm 5. Finally, in Section H, we present a table with main notations used in this work.

## A    Related Work

Sequential decision-making under uncertainty has been extensively studied in machine learning under the guise of multi-armed bandit (MAB) problems. In general, MAB problems can be split into two distinct flavors, which differ in the type of feedback agents receive. Crucially, in both problems the key is trading off exploration of actions and exploiting ones current knowledge.

In the first class of MAB problems, the stochastic MAB, playing an action results in an unbiased estimate of the utility of playing that action. Solutions to the problem can be split among two dominant algorithmic paradigms. The first, based on principle of optimism in the face of uncertainty encompasses the well known upper confidence bounds (UCB) algorithm Lattimore and Szepesvári (2020); Lai and Robbins (1985) and its variants, while the second, based on Thompson sampling takes a Bayesian approach Russo et al. (2017); Thompson (1933) Each of these approaches are known to have optimal performance measured in terms of *regret:* the expected cumulative utility generated from the algorithm's chosen actions compared to the expected utility that could have been generated from always choosing the best possible action (i.e., the best action that one would choose with full information) Lattimore and Szepesvári (2020); Agrawal and Goyal (2012). In particular, these algorithms are known to incur *logarithmic* regret, i.e., regret that grows at most logarithmically over time— which is known to be optimal for this class of problems up to constant factors. In our paper we present an algorithmic framework for learning in matching markets that works with either class of algorithm, and further incurs logarithmic regret *even* while dealing with competition.

The second class of multi-armed bandit problems, coming from the literature on learning in games, seeks algorithms that can perform against arbitrary feedback sequences Cesa-Bianchi and Lugosi (2006). Solutions to this class of problems, known as adversarial bandit algorithms, are an active research topic. While it is well known that using simple strategies like multiplicative weights can guarantee regret against the best fixed action in hindsight on the order of $\sqrt{T}$ against worst-case adversaries Cesa-Bianchi and Lugosi (2006), designing algorithms that can improve upon this when adversaries are *not* worst case remains an open research problem. In this paper we leverage advances on the development of *path-length* adversarial regret algorithms that address this problem and guarantee regret that directly depends on the amount of variation an adversary presents Bubeck et al. (2019); Wei and Luo (2018).

We briefly remark that there exists several lines of research on multi-agent bandits. One of them is on multi-agent bandits with collisions (with applications primarily in the area of spectrum sharing in wireless networksLiu and Zhao (2010); Kalathil et al. (2014); Rosenski et al. (2016); Lugosi and Mehrabian (2021); Bubeck et al. (2020)). In such models the arms do not have preferences and if more than one agents collide at any arm then no one receives any utility or attains maximum possible loss. However, these models differ from us since we consider that both sides of markets have preference over one another and when there is a collision only one agents gets matched. Another line of research deals with the problem of collaboratively learning an instance of multi-armed bandit Buccapatnam et al. (2015); Chakraborty et al. (2017); Sankararaman et al. (2019) where agents can communicate. Note that in these settings there is no competition that is more than one agents apply at same arm at same time.

The particular intersection of MABs and two-sided matching markets that we analyze has seen a flurry of recent works Liu et al. (2020, 2021); Basu et al. (2021); Sankararaman et al. (2021). To the best of our knowledge, Das and Kamenica (2005), presented the first numerical study on effectively using MAB algorithms to learn preferences in matching markets. However, it was only recently that Liu et al. (2020) rigorously formulated the bandit learning problem in the matching markets, and generalized the notion of *regret* from the MAB literature to matching markets in terms of *stable regret*— i.e., the expected cumulative utility benchmarked against the expected cumulative reward that would have been received if everyone in the market requested their match in a certain stable match[9]. Moreover, they proposed a *centralized* UCB-based algorithm that facilitates the matching between agents and firms given each agents' current beliefs over their preferences and history of play, while ensuring that $\mathcal{O}(|\mathcal{A}||\mathcal{F}|\log(T))$ regret for a UCB based algorithm, where $\mathcal{A}$ is the set of agents, $\mathcal{F}$ is the set of firms, and $T$ is the time horizon of the problem. In follow up work Liu et al. (2021) proposed a *decentralized* bandit learning algorithm based on UCB that allows each user to take its decision in a decentralized manner and still "converge" to stable matching while incurring $O(\exp(|\mathcal{F}|^4)\log^2(T))$ regret. More recently Kong et al. (2022) proposed a thompson sampling based variant of Liu et al. (2021). However, these algorithms requires the knowledge of outcomes at other firms at every round, leaving algorithms that are based solely on agents' own history of play as an open problem. Concurrently, Sankararaman et al. (2021) proposed an algorithm that works in phases and makes use of communication between agents to coordinate agents' actions. Under this information structure the algorithm achieves $\mathcal{O}\left(|\mathcal{F}|^2|\mathcal{A}|^2\log(T)\right)$ regret. Moreover their guarantees require that firms have homogeneous preference over the agents (also referred as *serial dictatorship*). Follow-up work, Basu et al. (2021) improved the regret for serial dictatorship to $\mathcal{O}\left(|\mathcal{F}||\mathcal{A}|\log(T)\right)$ by proposing a new algorithm. Additionally, they also showed that if the assumption of serial dictatorship is relaxed to a weaker structural condition then they obtain $O(poly(|\mathcal{A}|,|\mathcal{F}|)\log(T))$ regret. Even though the proposed algorithm in Basu et al. (2021) has decentralization it is a phase based algorithm, the agents act according to a coordinated protocol at some rounds. In this paper we propose a simple, decentralized, communication and coordination free algorithm in which agents make use of their own local information to learn while matching. Unlike previous works Liu et al. (2020, 2021); Sankararaman et al. (2021); Basu et al. (2021) where the algorithms are constructed using a UCB subroutine, we also show that our algorithmic design paradigm can be also seamlessly extended to Thompson sampling variant.

We would also like to remark about another line of research at the intersection of multiarmed bandits and matching markets Jagadeesan et al. (2021); Johari et al. (2016); Cen and Shah (2021) which consider the problem of learning preferences from the perspective of a platform.

# B  Adaptive Adversarial Algorithms

In this work we deploy the optimistic mirror descent based adversarial bandit module. We adapt algorithms from Bubeck et al. (2019), who improve the algorithm originally proposed in Wei and Luo (2018). In this section we recap the results from Bubeck et al. (2019). For the sake of completeness we restate the problem formulation and algorithm here. Towards the end we will specialize their results in the setting of this paper and state an useful result which presents the regret of such algorithms, in the context of the bandit structure described in Sec 3.2, in terms of the number of matchings and collisions.

## B.1  Problem formulation from Bubeck et al. (2019)

In this section we review algorithm described in Bubeck et al. (2019) which is an improvement over the one described in Wei and Luo (2018). Consider a multi-armed bandit problem that proceeds in $\tau$ time steps with $A \leq \tau$ fixed actions. In each round $t$, the algorithm selects one arm $i(t) \in [A]$ and simultaneously an adversary decides the loss vector $\ell(t) = (\ell_i(t))_{i \in [A]} \in [-1, 1]^A$. Note that the adversary can be an adaptive one in that it can base its actions on the past rounds of algorithm's actions. The goal of the algorithm is to minimize the gap between total accumulated loss and the loss

---

[9]Note that the stable matching need not be unique in general. Thus the stable regret has to be always specified with respect to which stable matching is being used. Typically, in literature two main stable matchings are considered namely *agent optimal stable matching* and *firm optimal stable matching*.

of best fixed arm in hindsight:

$$\text{Regret}^{(\text{adv})}(\tau) = \max_{i^\star \in [A]} \mathbb{E}\left[\sum_{t=1}^{\tau} \ell_{i(t)}(t) - \sum_{t=1}^{\tau} \ell_{i^\star}(t)\right].$$

The algorithm is based on the optimistic mirror descend framework. At any time $t$, the algorithm samples an arm $i(t) \in [A]$ with probability $p(t) \in \Delta([A])$. The algorithm only receives the loss for the action taken and not other actions. Therfore, upon receiving the loss $\ell_{i(t)}(t)$ the algorithm creates an unbiased estimator of losses for other actions. The estimator is

$$\hat{L}_i(t) = \frac{\ell_i(t) - L(t-1)}{2p_i(t)}\mathbb{1}\left(i(t) = i\right) + \frac{1 + L(t-1)}{2}, \quad \forall\, i$$

The unbiased loss estimate $\hat{L}(t)$ is used to update the an auxiliary probability distribution $x(t+1) \in \Delta([A])$ through an optimistic mirror descend update with learning rate $\eta$. The optimistic mirror descend update is constructed from the Bregman divergence[10] associated with a log-barrier regularizer $\mathbb{R}^A \ni x \mapsto \psi(x) = \frac{1}{\eta}\sum_{i=1}^{A}\ln\frac{1}{x_i}$ as follows

$$x(t+1) = \arg\min_{z \in \Delta([A])} \langle z, \hat{L}(t)\rangle + D_\psi(z, x(t)).$$

The distribution $x(t+1)$ is used to update the arm sampling distribution $p(t+1)$ after mixing a small bias towards most recently picked arm as follows

$$p(t+1) = (1 - \lambda(t+1))x(t+1) + \lambda(t+1)\mathbf{e}_{i(t)}$$

where $\mathbf{e}_{i^t} \in \mathbb{R}^A$ is an element of standard basis in $\mathbb{R}^A$ with $i(t)$ element as 1 and all others as zero and $\lambda(t+1) = \frac{\lambda(1-L(t))}{2+\lambda(1-L(t))}$ for some $\lambda > 0$.

---

**Algorithm 3:** Optimistic Mirror Descend based Adversarial Bandit Algorithm

---

**Parameters:** $\eta, \lambda \in (0,1), p(1), x(1) = \mathsf{Unif}([A]), \psi(x) = \frac{1}{\eta}\sum_{i=1}^{A}\ln\frac{1}{x_i}$

1 **for** $t = 1, 2, .., \tau$ **do**
2     Play $i(t) \sim p(t)$ and observe $L(t) = \ell_{i(t)}(t)$
3     Construct an unbiased estimator $\hat{L}_i(t) = \frac{\ell_i(t)-L(t-1)}{2p_i(t)}\mathbb{1}\left(i(t) = i\right) + \frac{1+L(t-1)}{2}$ for all $i \in [A]$
4     Update $x(t+1) = \arg\min_{z \in \Delta([A])}\langle z, \hat{L}(t)\rangle + D_\psi(z, x(t))$
5     $p(t+1) = (1 - \lambda(t+1))x(t+1) + \lambda(t+1)\mathbf{e}_{i(t)}$ where $\lambda(t+1) = \frac{\lambda(1-L(t))}{2+\lambda(1-L(t))}$
6 **end**

---

Against the preceding backdrop, we restate Theorem 2 from Bubeck et al. (2019) below:

**Theorem 9.** *Algorithm 3 with* $\eta \leq \frac{1}{50}$, $\lambda = 8\eta$ *ensures that*

$$\text{Regret}^{(\text{adv})}(\tau) = \mathcal{O}\left(\frac{A\ln(T)}{\eta}\right) + 8\eta\mathbb{E}\left[V(T)\right]$$

*where* $V(T) := \sum_{t=2}^{T}|\ell_{i(t-1)}(t) - \ell_{i(t-1)}(t-1)|$ *is commonly referred as "path-length".*

**Remark 10.** *Note that Theorem 2 in Bubeck et al. (2019) requires[11]. But in fact the proof goes through for* $\eta \leq 1/50$. $\eta \leq 1/162$ *and* $\lambda = 8\eta$. *This is because in Bubeck et al. (2019) for the proof of Theorem 2, they directly lift (Wei and Luo, 2018, Theorem 7) where* $\eta \leq 1/162$ *which is not tuned efficiently.*

---

[10]Bregman divergence between two point $x, y$ with respect to a convex regularizer $\psi$ is given as $D_\psi(x, y) = \psi(x) - \psi(y) - \langle \nabla\psi(y), x - y\rangle$.

[11]Moreover, it is an algebraic exercise to establish that $\eta < \frac{1}{24}$ and $\lambda = \frac{1-12\eta-c\cdot\sqrt{1-24\eta}}{24}$ also works for some $c \in (0,1)$. But we don't go in this direction to retain simplicity of algorithmic description.

## B.2 Adaptive Adversarial Module

In this section we describe AB_Subroutine in Algorithm 2 which is based on the algorithm presented in Sec B.1.

For any $(a, f) \in \mathcal{A} \times \mathcal{F}$, the adversarial bandit module associated with $(a, f)$ ( as described in Algorithm 4 ) is a version of Algorithm 3 for case when there are two actions: *request the firm $f$* or *prune the firm $f$*. In addition, the loss incurred due to pruning the firm $f$ is always 0 while the loss incurred due to pulling an firm $f$ depends on whether the agent $a$ got matched with it or collided with it. In this special case of two actions, the optimistic mirror descent update (line 4 in Algorithm 3) can be obtained in closed form (see Lemma 12). Note that the adversarial bandit module associated with any agent-firm tuple $(a, f)$ is only used when $t \in \tau_{a,f}(T) \subset [T]$.

**Lemma 11.** *Given a scalar $\eta \leq \frac{1}{50}$, for any agent-firm pair $(a, f) \in \mathcal{A} \times \mathcal{F}$, the regret of the adversarial bandit algorithm is bounded as*

$$\mathbb{E}[Regret_{a,f}^{(adv)}(\tau_{a,f}(T))] \leq \mathcal{O}\left(\frac{\log(T)}{\eta}\right) + 32\eta\mathbb{E}\left[\min\left\{M_{a,f}^\star(T), C_{a,f}^\star(T), M_{a,f}(T) + C_{a,f}(T)\right\}\right],$$

*where $M_{a,f}^\star(T) = \sum_{t=1}^{T} \mathbb{1}\left(H_{a,f}^{\mathsf{c}}(t)\right)$ and $C_{a,f}^\star(T) = \sum_{t=1}^{T} \mathbb{1}\left(H_{a,f}(t)\right)$.*

*Proof.* To prove this lemma we only need to bound the path length $V_{a,f}(T)$ in Theorem 9. We claim that the path length $V_{a,f}(T) \leq \min\{C_{a,f}^\star(T), M_{a,f}^\star(T)\}$. Recall $\tau_{a,f}(T) = \{t \in T : E_{a,f}^{(\mathsf{c})}(t) = 1\}$. For the remaining proof for any $t \in \tau_{a,f}(T)$ by $t - 1$ we mean $\max\{\mathfrak{t} < t : \mathfrak{t} \in \tau_{a,f}(T)\}$. For any $t \in \tau_{a,f}(T)$, let's denote the loss due to pruning at time $t$ by $\ell_{a,f}^{(prune)}(t)$ and similarly let the loss due to pulling at time $t$ by $\ell_{a,f}^{(pull)}(t)$. Note that by construction, the loss due to the pruning operation is deterministic and zero. That is, for any $t \in \tau_{a,f}(T)$, $\ell_{a,f}^{(prune)}(t) = 0$ and $\ell_{a,f}^{(pull)}(t) = 1 - 2Y_a(t)$. Furthermore, note that

$$\begin{aligned}
V_{a,f}(T) &\leq \sum_{t \in \tau_{a,f}(T)} |\ell_{a,f}^{(pull)}(t) - \ell_{a,f}^{(pull)}(t-1)| \\
&\underset{(a)}{\leq} 2 \sum_{t \in \tau_{a,f}(T)} \mathbb{1}\left(H_{a,f}(t-1), H_{a,f}^{\mathsf{c}}(t)\right) + \mathbb{1}\left(H_{a,f}^{\mathsf{c}}(t-1), H_{a,f}(t)\right) \\
&\leq 4 \min\left\{\sum_{t=1}^{T} \mathbb{1}\left(H_{a,f}^{\mathsf{c}}(t)\right), \sum_{t=1}^{T} \mathbb{1}\left(H_{a,f}(t)\right)\right\} \\
&= 4 \min\left\{M_{a,f}^\star(T), C_{a,f}^\star(T)\right\}
\end{aligned}$$

where the factor of 2 in is by the fact that a path length change in going from matching to potential collision or collision to potential matching is 2. The remaining inequalities follow from algebra.

Furthermore, we have

$$V_{a,f}(T) = \sum_{t\in\tau_{a,f}(T)} \mathbb{1}\left(P_{a,f}(t)=1, P_{a,f}(t-1)=1\right)|\ell_{a,f}^{(pull)}(t) - \ell_{a,f}^{(pull)}(t-1)|$$

$$+ \sum_{t\in\tau_{a,f}(T)} \mathbb{1}\left(P_{a,f}(t)=0, P_{a,f}(t-1)=1\right)|\ell_{a,f}^{(pull)}(t) - \ell_{a,f}^{(pull)}(t-1)|$$

$$\leq \sum_{t\in\tau_{a,f}(T)} \mathbb{1}\left(P_{a,f}(t)=1, P_{a,f}(t-1)=1\right)|\ell_{a,f}^{(pull)}(t) - \ell_{a,f}^{(pull)}(t-1)|$$

$$+ 2\sum_{t\in\tau_{a,f}(T)} \mathbb{1}\left(P_{a,f}(t)=0, P_{a,f}(t-1)=1\right)$$

$$= \sum_{t\in\tau_{a,f}(T)} \mathbb{1}\left(P_{a,f}(t)=1, P_{a,f}(t-1)=1\right)|\ell_{a,f}^{(pull)}(t) - \ell_{a,f}^{(pull)}(t-1)|$$

$$+ 2\sum_{t=2}^{T} \mathbb{1}\left(P_{a,f}(t)=0, P_{a,f}(t-1)=1\right)$$

$$= 2\sum_{t\in\tau_{a,f}(T)} \mathbb{1}\left(P_{a,f}(t)=1, P_{a,f}(t-1)=1, Y_a(t)=0, Y_a(t-1)=1\right)$$

$$+ 2\sum_{t\in\tau_{a,f}(T)} \mathbb{1}\left(P_{a,f}(t)=1, P_{a,f}(t-1)=1, Y_a(t)=1, Y_a(t-1)=0\right)$$

$$+ 2\sum_{t\in\tau_{a,f}(T)} \mathbb{1}\left(P_{a,f}(t)=0, P_{a,f}(t-1)=1\right)$$

$$\leq 2\left(\sum_{t\in\tau_{a,f}(T)} \mathbb{1}\left(P_{a,f}(t)=1, Y_a(t)=0\right) + \mathbb{1}\left(P_{a,f}(t-1)=1, Y_a(t-1)=1\right)\right)$$

$$+ 2\sum_{t\in\tau_{a,f}(T)} \mathbb{1}\left(P_{a,f}(t)=0, P_{a,f}(t-1)=1\right)$$

$$\leq 4\left(M_{a,f}(T) + C_{a,f}(T)\right)$$

### B.2.1 AB_Subroutine

---

**Algorithm 4:** AB_Subroutine

**Input**        : $P_{a,f}, x_{a,f}, p_{a,f}, L_{a,f}, Y_a$
**Parameters:** $\eta \leq \frac{1}{50}, \lambda = 8\eta$

1 **if** $P_{a,f} = 0$ **then**
2 $\quad$ Set $\hat{L}_{a,f}^{(\text{prune})} = \frac{-L_{a,f}}{2(1-p_{a,f})} + \frac{L_{a,f}+1}{2}, \quad \hat{L}_{a,f}^{(\text{pull})} = \frac{1+L_{a,f}}{2}$
3 $\quad$ Update $L_{a,f} \longleftarrow 0$
4 **end**
5 **if** $P_{a,f} = 1$ **then**
6 $\quad$ Set $\hat{L}_{a,f}^{(\text{prune})} = \frac{1+L_{a,f}}{2}, \quad \hat{L}_{a,f}^{(\text{pull})} = \frac{1-2Y_a-L_{a,f}}{2p_{a,f}} + \frac{1+L_{a,f}}{2}$
7 $\quad$ Update $L_{a,f} \longleftarrow 1 - 2Y_a$
8 **end**
9 Set $\xi = \eta\left(\hat{L}_{a,f}^{(\text{pull})} - \hat{L}_{a,f}^{(\text{prune})}\right) + \frac{1}{x_{a,f}} - \frac{1}{1-x_{a,f}}$
10 Update $x_{a,f} \longleftarrow \frac{2+\xi-\sqrt{4+\xi^2}}{2\xi}$ and set $\Lambda_{a,f} = \frac{\lambda(1-L_{a,f})}{2+\lambda(1-L_{a,f})}$ Update
$\quad p_{a,f} \longleftarrow (1-\Lambda_{a,f})x_{a,f} + \Lambda_{a,f}P_{a,f}$
**Output**        : $L_{a,f}, x_{a,f}, p_{a,f}$

---

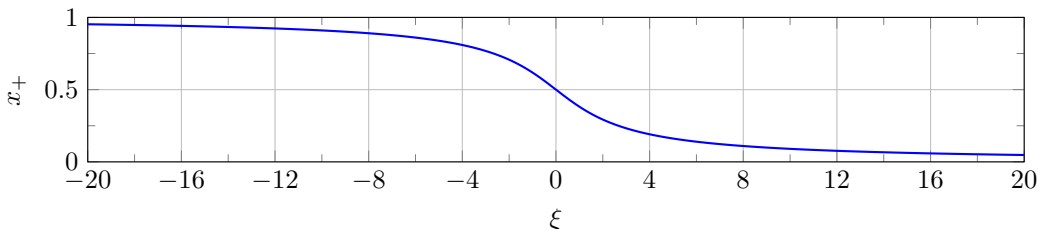

Figure 1: Update function of pulling probability based on line 10 in Algorithm 4

### B.3 Technical Lemma

**Lemma 12.** *For any $L \in \mathbb{R}^2$ and $X \in \Delta(\mathbb{R}^2)$ the update $X_+ = \arg\min_{Z \in \Delta(\mathbb{R}^2)} \langle Z, L \rangle + D_\psi(Z, X)$ can be analytically solved to be $X_+ = [x_+, 1 - x_+]$ where*

$$x_+ = \frac{2 + \xi - \sqrt{4 + \xi^2}}{2\xi} \tag{B.1}$$

*where $\xi = \eta(L_1 - L_2) + \frac{1}{X_1} - \frac{1}{X_2}$. For better interpretation we provide the graph for update (B.1) in the Figure 1.*

*Proof.* For any $X, Z \in \Delta(\mathbb{R}^2)$ we represent $X = [x, 1-x]$ and $Z = [z, 1-z]$ for $x, z \in [0, 1]$. Under this notation we can write $D_\psi(Z, X) = \frac{1}{\eta} \left( \log\left(\frac{x}{z}\right) + \log\left(\frac{1-x}{1-z}\right) + \frac{z-x}{x} + \frac{x-z}{1-x} \right)$. Thus the optimization problem becomes

$$x_+ = \arg\min_{z \in [0,1]} \langle z, L \rangle + D_\psi(z, X)$$

$$= \arg\min_{z \in [0,1]} zL_1 + (1-z)L_2 + \frac{1}{\eta} \left( \log\left(\frac{x}{z}\right) + \log\left(\frac{1-x}{1-z}\right) + \frac{z-x}{x} + \frac{x-z}{1-x} \right)$$

$$= \arg\min_{z \in [0,1]} zL_1 + (1-z)L_2 + \frac{1}{\eta} \left( -\log(z) - \log(1-z) + \frac{z}{x} - \frac{z}{1-x} \right)$$

Let $f(z) = zL_1 + (1-z)L_2 + \frac{1}{\eta}\left( -\log(z) - \log(1-z) + \frac{z}{x} - \frac{z}{1-x} \right)$. Note that $f(0) = +\infty$, and $f(1) = +\infty$ so the minimizer of $f(z)$ lies stricly inside $[0, 1]$. Therefore $\nabla f(x_+) = 0$. We compute

$$\nabla f(z) = L_1 - L_2 + \frac{1}{\eta(1-z)} - \frac{1}{\eta z} + \frac{1}{\eta x} - \frac{1}{\eta(1-x)} = L_1 - L_2 + \frac{2z-1}{\eta z(1-z)} + \frac{1}{\eta x} - \frac{1}{\eta(1-x)}$$

Imposing the condition $\nabla f(x_+) = 0$ implies that

$$\xi x_+^2 - (2+\xi)x_+ + 1 = 0$$

where $\xi = \eta(L_1 - L_2) + \frac{1}{x} - \frac{1}{1-x}$. Thus there are two possibilities

$$x_+ = \frac{2 + \xi + \sqrt{4 + \xi^2}}{2\xi}, \quad \text{or} \quad x_+ = \frac{2 + \xi - \sqrt{4 + \xi^2}}{2\xi},$$

However the first possibility implies that $x_+ > 1$, thus the only solution which lies in $(0, 1)$ is the latter. This completes the proof. $\qquad\square$

$\square$

## C   Proofs of main Lemmas

We introduce the following notation for every $a \in \mathcal{A}, f \in \mathcal{F}$

$$H_{a,f}(t) = \mathbb{1}\left( \exists a' \in \mathcal{A} : f_{a'}(t) = f, u_f(a') > u_f(a) \right),$$

which characterizes an event some agent more preferred than $a$ by firm $f$ has requested firm $f$. We now present the proofs of Lemmas in main paper in the following subsections.

## C.1 Proof of Lemma 8

Proof of Lemma 8 follows directly from the following Lemma.

**Lemma 13.** *The event that agent $a$ chooses a firm $f \in \mathcal{F}$ at time $t \in [T]$ satisfies*

$$\{Y_a(t) = 1, f_a(t) = f\} \subset \left\{Y_a(t) = 1, \mathsf{UCB}_{a,f_a^*}(t) \leq \mathsf{UCB}_{a,f}(t)\right\} \bigcup \left\{E_{a,f}^{(r)}(t) = 1, E_{a,f_a^*}^{(r)}(t) = 0\right\}. \tag{C.1}$$

*Proof.* For any agent $a$ fix some $f$. Recall that $f_a(t) = f$ implies that agent a has chosen to pull arm $f$. Based on design of Algorithm 2 there are two possibilities: either all the firms with higher UCB than firm $f$ got pruned and the firm $f$ was requested; or all of the firms in $\mathcal{F}$ got pruned and the firm $f$ got selected as it was having highest UCB. Thus,

$$\{f_a(t) = f\} = \left\{E_{a,f}^{(r)}(t) = 1\right\} \bigcup \left\{E_{a,f}^{(r)}(t) = 0 \; \forall \, f \in \mathcal{F}, \mathsf{UCB}_{a,f} \geq \mathsf{UCB}_{a,f'} \; \forall \, f' \in \mathcal{F}\right\}$$

$$\underset{(i)}{=} \left\{E_{a,f}^{(r)}(t) = 1, \mathsf{UCB}_{a,f_a^*}(t) \geq \mathsf{UCB}_{a,f}(t)\right\} \bigcup \left\{E_{a,f}^{(r)}(t) = 1, \mathsf{UCB}_{a,f_a^*}(t) \leq \mathsf{UCB}_{a,f}(t)\right\}$$

$$\bigcup \left\{E_{a,f}^{(r)}(t) = 0 \; \forall \, f \in \mathcal{F}, \mathsf{UCB}_{a,f} \geq \mathsf{UCB}_{a,f'} \; \forall \, f' \in \mathcal{F}\right\}$$

$$\underset{(ii)}{\subset} \left\{E_{a,f}^{(r)}(t) = 1, \mathsf{UCB}_{a,f_a^*}(t) \geq \mathsf{UCB}_{a,f}(t)\right\} \bigcup \left\{E_{a,f}^{(r)}(t) = 1, \mathsf{UCB}_{a,f_a^*}(t) \leq \mathsf{UCB}_{a,f}(t)\right\}$$

$$\bigcup \left\{\mathsf{UCB}_{a,f_a^*}(t) \leq \mathsf{UCB}_{a,f}(t)\right\}$$

$$\underset{(iii)}{\subset} \left\{E_{a,f}^{(r)}(t) = 1, E_{a,f_a^*}^{(r)}(t) = 0, \mathsf{UCB}_{a,f_a^*}(t) \geq \mathsf{UCB}_{a,f}(t)\right\}$$

$$\bigcup \left\{E_{a,f}^{(r)}(t) = 1, \mathsf{UCB}_{a,f_a^*}(t) \leq \mathsf{UCB}_{a,f}(t)\right\} \bigcup \left\{\mathsf{UCB}_{a,f_a^*}(t) \leq \mathsf{UCB}_{a,f}(t)\right\}$$

$$\underset{(iv)}{\subset} \left\{E_{a,f}^{(r)}(t) = 1, E_{a,f_a^*}^{(r)}(t) = 0, \mathsf{UCB}_{a,f_a^*}(t) \geq \mathsf{UCB}_{a,f}(t)\right\} \bigcup \left\{\mathsf{UCB}_{a,f_a^*}(t) \leq \mathsf{UCB}_{a,f}(t)\right\}$$

$$\underset{(v)}{\subset} \left\{E_{a,f}^{(r)}(t) = 1, E_{a,f_a^*}^{(r)}(t) = 0\right\} \bigcup \left\{\mathsf{UCB}_{a,f_a^*}(t) \leq \mathsf{UCB}_{a,f}(t)\right\}$$

where in $(i)$ we introduced two complementary events $\{\mathsf{UCB}_{a,f_a^*}(t) \geq \mathsf{UCB}_{a,f}(t)\}$ and $\{\mathsf{UCB}_{a,f_a^*}(t) \leq \mathsf{UCB}_{a,f}(t)\}$. Note that $(ii)$ holds due to the fact that $\{\mathsf{UCB}_{a,f_a(t)} \geq \mathsf{UCB}_{a,f} \; \forall \, f \in \mathcal{F}\}$ implies $\{\mathsf{UCB}_{a,f_a(t)} \geq \mathsf{UCB}_{a,f_a^*}\}$. Furthermore, $(iii)$ holds due to the fact that a firm with lower UCB will be pulled only if all the firms with higher UCB are pruned. Finally, $(iv), (v)$ holds by dropping appropriate events.

The result follows by noting that

$$\mathbb{1}\left(Y_a(t) = 1, f_a(t) = f\right)$$

$$\subset \left(\left\{E_{a,f}^{(r)}(t) = 1, E_{a,f_a^*}^{(r)}(t) = 0\right\} \bigcup \left\{\mathsf{UCB}_{a,f_a^*}(t) \leq \mathsf{UCB}_{a,f}(t)\right\}\right) \bigcap \mathbb{1}\left(Y_a(t) = 1\right)$$

$$\subset \left\{Y_a(t) = 1, \mathsf{UCB}_{a,f_a^*}(t) \leq \mathsf{UCB}_{a,f}(t)\right\} \bigcup \left\{E_{a,f}^{(r)}(t) = 1, E_{a,f_a^*}^{(r)}(t) = 0\right\}$$

$\square$

**Remark 14.** *The results in Lemma 13 holds even if we replace UCB subroutine in Algorithm 2 with any other index based stochastic bandit subroutine, e.g. Thompson sampling.*

## C.2 Proof of Lemma 7

We present the proof of each result **(L1)-(L5)** in Lemma 7 individually in the following subsubsections. Before that we define an important notation as follows:

$$H_{a,f}(t) = \mathbb{1}\left(\exists \, a' \in \mathcal{A} : f_{a'}(t) = f, u_f(a') \geq u_f(a)\right) \tag{C.2}$$

### C.2.1 Proof of (L1) in Lemma 7

From (4.1) we get

$$\sum_{i=1}^{k} \sum_{a \in \mathcal{A}_i} R_a \le \bar{\Delta} \sum_{i=1}^{k} \sum_{a \in \mathcal{A}_i} \sum_{f \in \mathbb{F}_a} \mathbb{E}[M_{a,f}(T)] + u \sum_{i=1}^{k} \sum_{a \in \mathcal{A}_i} \sum_{f \in F \setminus \{f_a^*\}} \mathbb{E}[C_{a,f}(T)]$$

$$+ \bar{u} \sum_{i=1}^{k} \sum_{a \in \mathcal{A}_i} \mathbb{E}[C_{a,f_a^*}(T)],$$

$$\le \bar{C} \Big( \sum_{i=1}^{k} \sum_{a \in \mathcal{A}_i} \sum_{f \in \mathbb{F}_a} \mathbb{E}[M_{a,f}(T)] + \sum_{i=1}^{k} \sum_{a \in \mathcal{A}_i} \sum_{f \in F \setminus \{f_a^*\}} \mathbb{E}[C_{a,f}(T)]$$

$$+ \sum_{i=1}^{k} \sum_{a \in \mathcal{A}_i} \mathbb{E}[\sum_{t=1}^{T} H_{a,f_a^*}(t)] \Big)$$

where $\bar{\Delta} = \max_{a,f} \Delta_a(f)$ and $\bar{u} = \max_a u_a(f_a^*)$. This completes the proof

### C.2.2 Proof of (L2) in Lemma 7

Proof of **(L2)** in Lemma 7 follows immediately from the following result.

**Lemma 15.** *For any agent $a \in \mathcal{A}$ using Algorithm 2 the expected number of matches with any set $\tilde{\mathcal{F}} \subset \mathbb{F}_a$ can be bounded as*

$$\mathbb{E}[M_{a,\tilde{\mathcal{F}}}(T)] \le \mathcal{O}\left( |\tilde{\mathcal{F}}| \left( \log(T) + \frac{\log(T)}{\Delta^2} \right) + \mathbb{E}\left[ \sum_{t=1}^{T} \mathbb{1}\left( H_{a,f_a^*}(t) \right) \right] \right)$$

*where $\Delta = \min_{a,f} \Delta_a(f)$.*

*Proof.* Note that we call an agent $a$ matches with firm $f$ at time $t$ if $Y_a(t) = 1$ and $f_a(t) = f$. Therefore the total number of matchings between $a$ and $f$ till time $T$ is $M_{a,f}(T) = \sum_{t=1}^{T} \mathbb{1}(Y_a(t) = 1, f_a(t) = f)$. Therefore from Lemma 8 the following holds for every $f \in \tilde{\mathcal{F}}$:

$$M_{a,\tilde{\mathcal{F}}}(T) = \sum_{f \in \tilde{\mathcal{F}}} \sum_{t=1}^{T} \mathbb{1}(Y_a(t) = 1, f_a(t) = f)$$

$$\le \sum_{f \in \tilde{\mathcal{F}}} \sum_{t=1}^{T} \left( \mathbb{1}\left( Y_a(t) = 1, f_a(t) = f, \mathsf{UCB}_{a,f}(t) \ge \mathsf{UCB}_{a,f_a^*}(t) \right) + \mathbb{1}\left( E_{a,f}^{(r)}(t) = 1, E_{a,f_a^*}^{(r)} = 0 \right) \right)$$

$$\le \sum_{f \in \tilde{\mathcal{F}}} \sum_{t=1}^{T} \mathbb{1}\left( Y_a(t) = 1, f_a(t) = f, \mathsf{UCB}_{a,f}(t) \ge \mathsf{UCB}_{a,f_a^*}(t) \right)$$

$$+ \sum_{t=1}^{T} \sum_{f \in \tilde{\mathcal{F}}} \mathbb{1}\left( E_{a,f}^{(r)}(t) = 1, E_{a,f_a^*}^{(r)} = 0 \right)$$

$$\le \underbrace{\sum_{f \in \tilde{\mathcal{F}}} \sum_{t=1}^{T} \mathbb{1}\left( Y_a(t) = 1, f_a(t) = f, \mathsf{UCB}_{a,f}(t) \ge \mathsf{UCB}_{a,f_a^*}(t) \right)}_{\text{Term A}} + \underbrace{\sum_{t=1}^{T} \mathbb{1}\left( E_{a,f_a^*}^{(r)} = 0 \right)}_{\text{Term B}}$$

For any fixed firm $f \in \tilde{\mathcal{F}}$ we now bound Term A. For that purpose, define an event

$$\mathcal{Z}_{a,f}(t) := \{\mathsf{UCB}_{a,f}(t) \ge u_a(f_a^*) - \epsilon\} = \left\{ \hat{\mu}_{a,f}(t-1) + \sqrt{\frac{2 \log(B_a(t))}{M_{a,f}(t-1)}} \ge u_a(f_a^*) - \epsilon \right\},$$

where $B_a(t) := 1 + \bar{M}_a(t) \log^2\left(\bar{M}_a(t)\right) \leq 1 + t \log^2(t) =: \bar{B}(t),$[12].

Using this notation, we have

$$\text{Term A} = \underbrace{\sum_{t=1}^{T} \mathbb{1}(Y_a(t) = 1, f_a(t) = f, \mathsf{UCB}_{a,f}(t) \geq \mathsf{UCB}_{a,f_a^*}(t), \mathcal{Z}_{a,f}(t))}_{\text{Term C}}$$

$$+ \underbrace{\sum_{t=1}^{T} \mathbb{1}(Y_a(t) = 1, f_a(t) = f, \mathsf{UCB}_{a,f}(t) \geq \mathsf{UCB}_{a,f_a^*}(t), \mathcal{Z}_{a,f}^{\mathsf{c}}(t))}_{\text{Term D}}$$

We shall first bound $\mathbb{E}[\text{Term C}]$ below:

$$\text{Term C} = \sum_{t=1}^{T} \mathbb{1}(Y_a(t) = 1, f_a(t) = f, \mathsf{UCB}_{a,f}(t) \geq \mathsf{UCB}_{a,f_a^*}(t), \mathcal{Z}_{a,f}(t))$$

$$\leq \sum_{t=1}^{T} \mathbb{1}(Y_a(t) = 1, f_a(t) = f, \mathcal{Z}_{a,f}(t))$$

$$= \sum_{t=1}^{T} \mathbb{1}\left(Y_a(t) = 1, f_a(t) = f, \hat{\mu}_{a,f}(t-1) + \sqrt{\frac{2 \log(B_a(t))}{M_{a,f}(t-1)}} \geq u_a(f_a^*) - \epsilon\right)$$

$$\leq \sum_{t=1}^{T} \mathbb{1}\left(Y_a(t) = 1, f_a(t) = f, \hat{\mu}_{a,f}(t-1) + \sqrt{\frac{2 \log(B_a(T))}{M_{a,f}(t-1)}} \geq u_a(f_a^*) - \epsilon\right)$$

$$= \sum_{t=1}^{T} \sum_{s=0}^{t-1} \mathbb{1}\left(Y_a(t) = 1, f_a(t) = f, \hat{\mu}_{a,f}^{(s)} + \sqrt{\frac{2 \log(B_a(T))}{s}} \geq u_a(f_a^*) - \epsilon, M_{a,f}(t-1) = s\right)$$

$$\leq \sum_{s=0}^{T-1} \sum_{t=s+1}^{T} \mathbb{1}\left(f_a(t) = f, \hat{\mu}_{a,f}^{(s)} + \sqrt{\frac{2 \log(B_a(T))}{s}} \geq u_a(f_a^*) - \epsilon, M_{a,f}(t-1) = s, M_{a,f}(t) = s+1\right)$$

$$\leq \sum_{s=0}^{T-1} \mathbb{1}\left(\hat{\mu}_{a,f}^{(s)} + \sqrt{\frac{2 \log(B_a(T))}{s}} \geq u_a(f_a^*) - \epsilon\right)$$

$$\leq \sum_{s=0}^{T-1} \mathbb{1}\left(\hat{\mu}_{a,f}^{(s)} - u_a(f) + \sqrt{\frac{2 \log(\bar{B}(T))}{s}} \geq \underbrace{u_a(f_a^*) - u_a(f)}_{\Delta_a(f)} - \epsilon\right),$$

where $\mu_{a,f}^{(s)}$ is defined to be the empirical utility that agent $a$ obtains on $s$ independent successful pulls of arm $f$. Using Lemma 19 to further bound $\mathbb{E}[\text{Term C}]$ we get

$$\mathbb{E}[\text{Term C}] \leq 1 + \frac{2}{(\Delta_a(f) - \epsilon)^2}\left(\log(\bar{B}(T) + \sqrt{\pi \log(\bar{B}(T))} + 1)\right)$$

---

[12] The inequality holds due to the fact that $\bar{M}_a(t) \leq t$ and monotonicity of the mapping $x \mapsto 1 + x \log^2(x)$.

Next, we bound $\mathbb{E}[\text{Term D}]$ below:

$$\mathbb{E}[\text{Term D}] = \mathbb{E}\left[\sum_{t=1}^{T} \mathbb{1}(Y_a(t) = 1, f_a(t) = f, \mathsf{UCB}_{a,f}(t) \geq \mathsf{UCB}_{a,f_a^*}(t), \mathsf{UCB}_{a,f}(t) \leq u_a(f_a^*) - \epsilon\right]$$

$$\leq \mathbb{E}\left[\sum_{t=1}^{T} \mathbb{1}\left(Y_a(t) = 1, \hat{\mu}_{a,f_a^*}(t-1) + \sqrt{\frac{2\log(B_a(t))}{M_{a,f_a^*}(t-1)}} \leq u_a(f_a^*) - \epsilon\right)\right]$$

$$\leq \sum_{t=1}^{T}\sum_{s=0}^{T-1} \mathsf{Pr}\left(\hat{\mu}_{a,f_a^*}^{(s)} + \sqrt{\frac{2\log(\bar{B}(t))}{s}} \leq u_a(f_a^*) - \epsilon\right)$$

$$\leq \sum_{t=1}^{T}\sum_{s=0}^{T-1} \exp\left(-\frac{s\left(\sqrt{\frac{2\log(\bar{B}(t))}{s}} + \epsilon\right)^2}{2}\right)$$

$$\leq \sum_{t=1}^{T} \frac{1}{\bar{B}(t)} \sum_{s=1}^{T} \exp\left(-\frac{s\epsilon^2}{2}\right)$$

$$\leq \frac{\epsilon^2}{2} \sum_{t=0}^{T-1} \frac{1}{\bar{B}(t)}$$

which can further be bounded as $\mathbb{E}[\text{Term D}] \leq \frac{5}{\epsilon^2}$ in (Lattimore and Szepesvári, 2020, Exercise 8.1). For simplicity we choose $\epsilon = \Delta_a(f)/2$ which ensures that $\mathbb{E}[\text{Term A}] \leq \mathcal{O}\left(\frac{\log(T)}{(\Delta_a(f))^2}\right)$

Now let's turn our attention to Term B which characterizes the number of times agent $a$ has pruned the stable match. Using Lemma 21 we have

$$\mathbb{E}[\text{Term B}] \leq \mathcal{O}\left(\mathbb{E}\left[\sum_{t=1}^{T} \mathbb{1}\left(H_{a,f_a^*}(t)\right)\right] + \mathcal{O}(\log(T))\right)$$

Thus the Term A is bounded by number of there can be potential collisions at the stable firm. This concludes the proof of this lemma. $\qquad\square$

### C.2.3 Proof of (L3) in Lemma 7

In this part, we prove a result which is more general than **(L3)** in Lemma 7.

**Lemma 16.** *Expected number of collisions faced by agent $a$ on the set of firms $\mathcal{F}^{\dagger} \subseteq \mathcal{F}\backslash\{f_a^*\}$*

$$\sum_{f \in \mathcal{F}^{\dagger}} \mathbb{E}[C_{a,f}(T)] \leq \mathcal{O}\left(|\mathcal{F}^{\dagger}|\log(T) + \mathbb{E}[M_{a,\underline{\mathcal{F}}_a^{\dagger}}(T)] + \mathbb{E}[M_{a,\bar{\mathcal{F}}_a^{\dagger}}(T)] + \mathbb{E}\left[\sum_{t=1}^{T} \mathbb{1}\left(H_{a,f_a^*}(t)\right)\right]\right), \tag{C.3}$$

*where $\underline{\mathcal{F}}_a^{\dagger} = \underline{\mathbb{F}}_a \cap \mathcal{F}^{\dagger}$ and $\bar{\mathcal{F}}_a^{\dagger} = \bar{\mathbb{F}}_a \cap \mathcal{F}^{\dagger}$. Additionally*

$$\mathbb{E}\left[C_{a,f_a^*}(T)\right] \leq \mathbb{E}\left[\sum_{t=1}^{T} \mathbb{1}\left(H_{a,f_a^*}(t)\right)\right] \tag{C.4}$$

*Proof.* To compute the number of collisions, we compute the following for $a \in \mathcal{A}$ and $f \in \mathcal{F} \backslash \{f_a^*\}$

$$\sum_{f \in \mathcal{F}^\dagger} C_{a,f}(T) = \sum_{f \in \mathcal{F}^\dagger} \sum_{t=1}^{T} \mathbb{1} \left( f_a(t) = f, H_{a,f}(t) \right)$$

$$= \sum_{f \in \mathcal{F}^\dagger} \sum_{t=1}^{T} \mathbb{1} \left( E_{a,f}^{(\mathrm{r})}(t) = 1, E_{a,f}^{(\mathrm{c})}(t) = 1, H_{a,f}(t) \right)$$

$$+ \sum_{f \in \mathcal{F}^\dagger} \sum_{t=1}^{T} \mathbb{1} \left( E_{a,f'}^{(\mathrm{r})}(t) = 0 \ \forall \ f' \in \mathcal{F}, f_a(t) = f, H_{a,f}(t) \right)$$

$$\leq \sum_{f \in \mathcal{F}^\dagger} \sum_{t=1}^{T} \mathbb{1} \left( E_{a,f}^{(\mathrm{r})}(t) = 1, E_{a,f}^{(\mathrm{c})}(t) = 1, H_{a,f}(t) \right) + \sum_{f \in \mathcal{F}^\dagger} \sum_{t=1}^{T} \mathbb{1} \left( E_{a,f_a^*}^{(\mathrm{r})}(t) = 0, f_a(t) = f \right),$$

$$\leq \sum_{f \in \mathcal{F}^\dagger} \sum_{t=1}^{T} \mathbb{1} \left( E_{a,f}^{(\mathrm{r})}(t) = 1, E_{a,f}^{(\mathrm{c})}(t) = 1, H_{a,f}(t) \right) + \sum_{t=1}^{T} \mathbb{1} \left( E_{a,f_a^*}^{(\mathrm{r})}(t) = 0 \right),$$

where the first inequality holds because $\{E_{a,f'}^{(\mathrm{r})}(t) = 0 \ \forall \ f' \in \mathcal{F}\}$ implies that $\{E_{a,f_a^*}^{(\mathrm{r})}(t) = 0\}$. Using (E.1) we have: for all $a \in \mathcal{A}$, $f \in \mathcal{F}$ and $\varpi \in (0, 32\eta) \subset (0, 1)$

$$\sum_{f \in \mathcal{F}^\dagger} \mathbb{E}[C_{a,f}(T)] \leq \sum_{f \in \mathcal{F}^\dagger} \left( (1 + \varpi) \mathbb{E}[M_{a,f}(T)] + \mathcal{O}(\log(T)) + \varpi \mathbb{E}[C_{a,f}(T)] + \mathbb{E}\left[ \sum_{t=1}^{T} \mathbb{1} \left( E_{a,f_a^*}^{(\mathrm{r})} = 0 \right) \right] \right)$$

$$\leq \mathcal{O} \left( |\mathcal{F}^\dagger| \log(T) + \sum_{f \in \mathcal{F}^\dagger} \mathbb{E}[M_{a,f}(T)] \right) + \mathbb{E}\left[ \sum_{t=1}^{T} \mathbb{1} \left( H_{a,f_a^*}(t) \right) \right] + \varpi \sum_{f \in \mathcal{F}^\dagger} \mathbb{E}[C_{a,f}(T)]$$

where the last inequality is due to Lemma 21. In summary,

$$\sum_{f \in \mathcal{F}^\dagger} \mathbb{E}[C_{a,f}(T)] \leq \mathcal{O} \left( |\mathcal{F}| \mathcal{O}(\log(T)) + \sum_{f \in \mathcal{F}^\dagger} \left( \mathbb{E}[M_{a,f}(T)] \right) \right) + \mathbb{E}\left[ \sum_{t=1}^{T} \mathbb{1} \left( H_{a,f_a^*}(t) \right) \right]$$

$$\leq \mathcal{O} \left( |\mathcal{F}^\dagger| \log(T) + \mathbb{E}[M_{a,\underline{\mathcal{F}}_a^\dagger}(T)] + \mathbb{E}[M_{a,\bar{\mathcal{F}}_a^\dagger}(T)] + \mathbb{E}\left[ \sum_{t=1}^{T} \mathbb{1} \left( H_{a,f_a^*}(t) \right) \right] \right)$$

This completes the proof of (C.3). We now prove (C.4). We note that

$$\mathbb{E}\left[ C_{a,f_a^*}(T) \right] = \mathbb{E}\left[ \sum_{t=1}^{T} \mathbb{1} \left( f_a(t) = f, H_{a,f_a^*}(t) \right) \right]$$

$$\leq \mathbb{E}\left[ \sum_{t=1}^{T} \mathbb{1} \left( H_{a,f_a^*}(t) \right) \right].$$

This completes the proof. $\qquad \square$

### C.2.4 Proof of (L4) in Lemma 7

We restate **(L4)** from Lemma 7 below:

**Lemma 17.** *For any $i \in [K]$ we have*

$$\sum_{j=1}^{i} \sum_{a \in \mathcal{A}_j} \mathbb{E}\left[ \sum_{t=1}^{T} \mathbb{1} \left( H_{a,f_a^*}(t) \right) \right] = \mathcal{O} \left( C_i |\mathcal{F}| \left( \sum_{j=1}^{i} |\mathcal{A}_j| \right) \log(T) \left( 1 + \frac{1}{\Delta^2} \right) \right),$$

*where $C_i$ is a constant dependent on market $\mathcal{M}_i$ such that $C_1 < C_2 < ... < C_K$.*

*Proof.* For any $k \in [K]$ define $S_k = \sum_{i=1}^{k} \sum_{a \in \mathcal{A}_i} \mathbb{E}[\sum_{t=1}^{T} \mathbb{1}(H_{a,f_a^*}(t))]$ and $Z(T,\Delta) = |\mathcal{F}| \log(T) (1 + \frac{1}{\Delta^2})$. Define $f(\theta; \ell) = \sum_{j=1}^{\ell} \theta^j$ and $f(\theta; 0) = 1$ and $g(\theta; \ell) = \sum_{j=0}^{\ell-1} \theta^j$. Moreover, let $\mathcal{H}_i = \sum_{a \in \mathcal{A}_i} \mathbb{E}[\sum_{t=1}^{T} \mathbb{1}(H_{a,f_a^*}(t))]$. Consequently $S_k = \sum_{i=1}^{k} \mathcal{H}_i$

We claim that

$$S_K \leq S_{K-\ell} + f(\theta; \ell)\mathcal{H}_{K-\ell} + \sum_{p=1}^{\ell} g(\theta; p) \sum_{a \in \mathcal{A}_{K-p+1}} \sum_{a' \in \cup_{j=1}^{K-\ell-1} \mathcal{A}_j} \mathbb{E}\left[M_{a',f_a^*}(T)\right] + Z(T,\Delta) \sum_{r=1}^{\ell} f(\theta; r)|\mathcal{A}_{K-r}| \tag{C.5}$$

We prove this via induction. We first show that this holds for $\ell = 1$. Indeed note that

$$S_K = S_{K-1} + \mathcal{H}_K = S_{K-1} + \sum_{a \in \mathcal{A}_K} \mathbb{E}\left[\sum_{t=1}^{T} \mathbb{1}(H_{a,f_a^*}(t))\right]$$

$$\underset{(a)}{\leq} S_{K-1} + \sum_{a \in \mathcal{A}_K} \sum_{a' \in \cup_{j=1}^{K-2} \mathcal{A}_j} \mathbb{E}\left[M_{a',f_a^*}(T)\right] + \sum_{a \in \mathcal{A}_K} \sum_{a' \in \mathcal{A}_{K-1}} \mathbb{E}\left[M_{a',f_a^*}(T)\right]$$

$$\underset{(b)}{=} S_{K-1} + \sum_{a \in \mathcal{A}_K} \sum_{a' \in \cup_{j=1}^{K-2} \mathcal{A}_j} \mathbb{E}\left[M_{a',f_a^*}(T)\right] + \sum_{a' \in \mathcal{A}_{K-1}} \sum_{f \in \mathcal{F}_K} \mathbb{E}\left[M_{a',f}(T)\right]$$

$$\underset{(c)}{\leq} S_{K-1} + \sum_{a \in \mathcal{A}_K} \sum_{a' \in \cup_{j=1}^{K-2} \mathcal{A}_j} \mathbb{E}\left[M_{a',f_a^*}(T)\right] + \sum_{a' \in \mathcal{A}_{K-1}} \mathbb{E}\left[M_{a',\mathbb{F}_{a'}}(T)\right]$$

$$\underset{(d)}{\leq} S_{K-1} + \theta \sum_{a' \in \mathcal{A}_{K-1}} \mathbb{E}\left[\sum_{t=1}^{T} \mathbb{1}(H_{a',f_{a'}^*}(t))\right] + \sum_{a \in \mathcal{A}_K} \sum_{a' \in \cup_{j=1}^{K-2} \mathcal{A}_j} \mathbb{E}\left[M_{a',f_a^*}(T)\right] + \theta|\mathcal{A}_{K-1}|Z(T,\Delta)$$

$$= S_{K-1} + \theta\mathcal{H}_{K-1} + \sum_{a \in \mathcal{A}_K} \sum_{a' \in \cup_{j=1}^{K-2} \mathcal{A}_j} \mathbb{E}\left[M_{a',f_a^*}(T)\right] + \theta|\mathcal{A}_{K-1}|Z(T,\Delta)$$

where the (a) holds due to $\alpha-$reducible structure which says that any agent in $\mathcal{A}_K$ will only get collided at stable arm if some agent from $\cup_{j=1}^{k-1}\mathcal{A}_j$ has also requested the stable firm. Next, $(b)$ holds due to the fact that for any agent $a \in \mathcal{A}_k$, the corresponding stable match $f_a^* \in \mathcal{F}_k$(see Remark 3). Next, (c) follows because for agents in $\mathcal{A}_{K-1}$, the set of suboptimal firms is super set of $\mathcal{F}_K$. This is again a property of $\alpha-$reducible structure. Finally $(d)$ follows from **(L2)** in Lemma 7 where $\theta$ is the corresponding constant from big-oh notation.

Suppose the bound in (C.5) holds for $\ell = L$ for some integer $\ell \in \{2, 3, ..., K\}$. Then we show it also holds for $\ell + 1$. That is,

$$S_K \underset{(a)}{\leq} S_{K-\ell} + f(\theta; \ell)\mathcal{H}_{K-\ell} + \sum_{p=1}^{\ell} g(\theta; p) \sum_{a \in \mathcal{A}_{K-p+1}} \sum_{a' \in \cup_{j=1}^{K-\ell-1} \mathcal{A}_j} \mathbb{E}\left[M_{a',f_a^*}(T)\right] + Z(T,\Delta) \sum_{r=1}^{\ell} f(\theta; r)|\mathcal{A}_{K-r}|$$

$$\underset{(b)}{=} S_{K-\ell-1} + g(\theta; \ell+1)\mathcal{H}_{K-\ell} + \sum_{p=1}^{\ell} g(\theta; p) \sum_{a \in \mathcal{A}_{K-p+1}} \sum_{a' \in \cup_{j=1}^{K-\ell-1} \mathcal{A}_j} \mathbb{E}\left[M_{a',f_a^*}(T)\right]$$

$$+ Z(T,\Delta) \sum_{r=1}^{\ell} f(\theta; r)|\mathcal{A}_{K-r}|$$

$$\underset{(c)}{\leq} S_{K-\ell-1} + g(\theta; \ell+1)\left(\mathcal{H}_{K-\ell} + \sum_{p=1}^{\ell} \sum_{a \in \mathcal{A}_{K-p+1}} \sum_{a' \in \mathcal{A}_{K-\ell-1}} \mathbb{E}\left[M_{a',f_a^*}(T)\right]\right)$$

$$+ \sum_{p=1}^{\ell} g(\theta; p) \sum_{a \in \mathcal{A}_{K-p+1}} \sum_{a' \in \cup_{j=1}^{K-\ell-2} \mathcal{A}_j} \mathbb{E}\left[M_{a',f_a^*}(T)\right] + Z(T,\Delta) \sum_{r=1}^{\ell} f(\theta; r)|\mathcal{A}_{K-r}|$$

$$\underset{(d)}{\leq} S_{K-\ell-1} + g(\theta;\ell+1)\left(\sum_{p=1}^{K-\ell-1}\sum_{a'\in\mathcal{A}_p}\sum_{a\in\mathcal{A}_{K-\ell}}\mathbb{E}[M_{a',f_a^*}] + \sum_{p=1}^{\ell}\sum_{a\in\mathcal{A}_{K-p+1}}\sum_{a'\in\mathcal{A}_{K-\ell-1}}\mathbb{E}\left[M_{a',f_a^*}(T)\right]\right)$$

$$+ \sum_{p=1}^{\ell}g(\theta;p)\sum_{a\in\mathcal{A}_{K-p+1}}\sum_{a'\in\cup_{j=1}^{K-\ell-2}\mathcal{A}_j}\mathbb{E}\left[M_{a',f_a^*}(T)\right] + Z(T,\Delta)\sum_{r=1}^{\ell}f(\theta;r)|\mathcal{A}_{K-r}|$$

$$\underset{(e)}{=} S_{K-\ell-1} + g(\theta;\ell+1)\left(\sum_{p=1}^{K-\ell-2}\sum_{a'\in\mathcal{A}_p}\sum_{a\in\mathcal{A}_{K-\ell}}\mathbb{E}[M_{a',f_a^*}] + \sum_{p=1}^{\ell+1}\sum_{a\in\mathcal{A}_{K-p+1}}\sum_{a'\in\mathcal{A}_{K-\ell-1}}\mathbb{E}\left[M_{a',f_a^*}(T)\right]\right)$$

$$+ \sum_{p=1}^{\ell}g(\theta;p)\sum_{a\in\mathcal{A}_{K-p+1}}\sum_{a'\in\cup_{j=1}^{K-\ell-2}\mathcal{A}_j}\mathbb{E}\left[M_{a',f_a^*}(T)\right] + Z(T,\Delta)\sum_{r=1}^{\ell}f(\theta;r)|\mathcal{A}_{K-r}|$$

$$\underset{(f)}{\leq} S_{K-\ell-1} + g(\theta;\ell+1)\left(\sum_{p=1}^{K-\ell-2}\sum_{a'\in\mathcal{A}_p}\sum_{a\in\mathcal{A}_{K-\ell}}\mathbb{E}[M_{a',f_a^*}] + \sum_{a'\in\mathcal{A}_{K-\ell-1}}\mathbb{E}\left[M_{a',\mathbb{F}_{a'}}(T)\right]\right)$$

$$+ \sum_{p=1}^{\ell}g(\theta;p)\sum_{a\in\mathcal{A}_{K-p+1}}\sum_{a'\in\cup_{j=1}^{K-\ell-2}\mathcal{A}_j}\mathbb{E}\left[M_{a',f_a^*}(T)\right] + Z(T,\Delta)\sum_{r=1}^{\ell}f(\theta;r)|\mathcal{A}_{K-r}|$$

$$\underset{(g)}{=} S_{K-\ell-1} + g(\theta;\ell+1)\left(\sum_{a'\in\mathcal{A}_{K-\ell-1}}\mathbb{E}\left[M_{a',\mathbb{F}_{a'}}(T)\right]\right)$$

$$+ \sum_{p=1}^{\ell+1}g(\theta;p)\sum_{a\in\mathcal{A}_{K-p+1}}\sum_{a'\in\cup_{j=1}^{K-\ell-2}\mathcal{A}_j}\mathbb{E}\left[M_{a',f_a^*}(T)\right] + Z(T,\Delta)\sum_{r=1}^{\ell}f(\theta;r)|\mathcal{A}_{K-r}|$$

$$\underset{(h)}{\leq} S_{K-\ell-1} + g(\theta;\ell+1)\left(\theta|\mathcal{F}|Z(T,\Delta)|\mathcal{A}_{K-\ell-1}| + \theta\mathcal{H}_{K-\ell-1}\right)$$

$$+ \sum_{p=1}^{\ell+1}g(\theta;p)\sum_{a\in\mathcal{A}_{K-p+1}}\sum_{a'\in\cup_{j=1}^{K-\ell-2}\mathcal{A}_j}\mathbb{E}\left[M_{a',f_a^*}(T)\right] + Z(T,\Delta)\sum_{r=1}^{\ell}f(\theta;r)|\mathcal{A}_{K-r}|$$

$$\underset{(i)}{=} S_{K-\ell-1} + f(\theta;\ell+1)\mathcal{H}_{K-\ell-1} + \sum_{p=1}^{\ell+1}g(\theta;p)\sum_{a\in\mathcal{A}_{K-p+1}}\sum_{a'\in\cup_{j=1}^{K-\ell-2}\mathcal{A}_j}\mathbb{E}\left[M_{a',f_a^*}(T)\right]$$

$$+ Z(T,\Delta)\sum_{r=1}^{\ell+1}f(\theta;r)|\mathcal{A}_{K-r}|$$

where $(a)$ holds by induction hypothesis, $(b)$ holds by definition of $S_k$ and $f(\theta;\ell), g(\theta;\ell)$, $(c)$ holds by moving some terms around and noting that $g(\theta;\cdot)$ is increasing. Next, $(d)$ holds by $\alpha-$reducbility and definition of $\mathcal{H}_k$ (same analysis as in base case of induction). Next, $(e)$ holds by splitting the terms. Next, $(f)$ holds by $\alpha-$reducilibility definition. Next $(g)$ holds by combining similar terms. Next $(h)$ holds by (**L2**) in Lemma 7. Next, $(i)$ holds due to combining similar terms.

Thus we conclude that induction claim (C.5) holds true. We know that $S_1 = 0$ therefore from (C.5) we obtain

$$S_k \leq Z(T,\Delta)\sum_{r=1}^{K-1}f(\theta;r)|\mathcal{A}_{K-r}| \tag{C.6}$$

$$\leq \left(\sum_{j=1}^{K-1}|\mathcal{A}_j|\right)K\theta^{K-1}Z(T,\Delta). \tag{C.7}$$

The term $C_k = k\theta^{k-1}$ in the statement. This completes the proof.

$\square$

### C.2.5 Proof of (L5) in Lemma 7

So only thing to bound is matching with superoptimal firms.

**Lemma 18.** *For any $k \in [K]$ we have*

$$\sum_{j=1}^{k} \sum_{a \in \mathcal{A}_j} \sum_{f \in \bar{\mathbb{F}}_a} \mathbb{E}[M_{a,f}(T)] \leq \mathcal{O}\left( C_i \left( \sum_{j=1}^{k-1} |\mathcal{A}_j| \right) |\mathcal{F}| \log(T) \left( 1 + \frac{1}{\Delta^2} \right) \right),$$

*where $C_i$ is a constant dependent on market $\mathcal{M}_i$ such that $C_1 < C_2 < ... < C_K$.*

*Proof.* For any $k \in [K]$, define $\tilde{S}_k = \sum_{i=1}^{k} \sum_{a \in \mathcal{A}_i} \mathbb{E}[M_{a,\bar{\mathbb{F}}_a}(T)]$ and $Z(T,\Delta) = |F| \log(T) \left( 1 + 1/\Delta^2 \right)$. Define $f(\theta; \ell) = \sum_{j=1}^{\ell} \theta^j$ and $f(\theta; 0) = 1$ and $g(\theta; \ell) = \sum_{j=0}^{\ell-1} \theta^j$. Moreover, let $\mathcal{H}_i = \sum_{a \in \mathcal{A}_i} \mathbb{E}[\sum_{t=1}^{T} \mathbb{1}\left(H_{a,f_a^*}(t)\right)]$. Let $\mathbb{M}_i = \sum_{a \in \mathcal{A}_i} \mathbb{E}[M_{a,\bar{\mathbb{F}}_a}(T)]$ then $\tilde{S}_k = \sum_{i=1}^{k} \mathbb{M}_i$. We claim that

$$\tilde{S}_k \leq \mathcal{O}\left( \tilde{\theta}^{k-1} \left( \sum_{j=1}^{k-1} |\mathcal{A}_j| \right) |\mathcal{F}| Z(T,\Delta) \right) \tag{C.8}$$

where $\tilde{\theta}$ is a constant greater than 1. Note that the bound holds for $k = 1$ as there is not super-optimal firms for those agents. Let (C.8) holds till some integer $K - 1$ then we show that it holds for $K$ as well. Indeed,

We claim that

$$\tilde{S}_K \leq \tilde{S}_{K-\ell} + f(\tilde{\theta}; \ell)\mathbb{M}_{K-\ell} + \sum_{p=1}^{\ell} g(\tilde{\theta}; p) \sum_{a \in \mathcal{A}_{K-p+1}} \sum_{f \in \cup_{j \leq K-\ell-1} \mathcal{F}_j} \mathbb{E}\left[M_{a,f}\right] + \sum_{p=1}^{\ell} f(\tilde{\theta}, p)\mathcal{H}_{K-p}$$

$$+ Z(T,\Delta) \sum_{p=1}^{\ell} f(\tilde{\theta}, p)|\mathcal{A}_{K-p}| \tag{C.9}$$

We prove (C.8) by induction. First, consider the case $\ell = 1$

$$\tilde{S}_K = \sum_{i=1}^{K} \sum_{a \in \mathcal{A}_i} \mathbb{E}[M_{a,\bar{\mathbb{F}}_a}(T)]$$

$$\underset{(a)}{=} \tilde{S}_{K-1} + \sum_{a \in \mathcal{A}_K} \mathbb{E}[M_{a,\bar{\mathbb{F}}_a}(T)]$$

$$\underset{(b)}{\leq} \tilde{S}_{K-1} + \sum_{a \in \mathcal{A}_K} \sum_{f \in \cup_{j \leq K-2} \mathcal{F}_j} \mathbb{E}[M_{a,f}(T)] + \sum_{a \in \mathcal{A}_K} \sum_{f \in \mathcal{F}_{K-1}} \mathbb{E}[M_{a,f}(T)]$$

$$\underset{(c)}{=} \tilde{S}_{K-1} + \sum_{a \in \mathcal{A}_K} \sum_{f \in \cup_{j \leq K-2} \mathcal{F}_j} \mathbb{E}[M_{a,f}(T)] + \sum_{a' \in \mathcal{A}_{K-1}} \sum_{a \in \mathcal{A}_K} \mathbb{E}[M_{a,f_{a'}^*}(T)]$$

$$\underset{(d)}{\leq} \tilde{S}_{K-1} + \tilde{\theta} \sum_{a' \in \mathcal{A}_{K-1}} \mathbb{E}[M_{a',\bar{\mathbb{F}}_{a'}}(T)] + \sum_{a \in \mathcal{A}_K} \sum_{a' \in \cup_{j \leq K-2} \mathcal{A}_j} \mathbb{E}[M_{a,f_{a'}^*}(T)] + \sum_{a' \in \mathcal{A}_{K-1}} \tilde{\theta}\left( H_{a',f_{a'}^*} + Z(T,\Delta) \right)$$

$$\underset{(e)}{=} \tilde{S}_{K-1} + \tilde{\theta}\mathbb{M}_{K-1} + \sum_{a \in \mathcal{A}_K} \sum_{f \in \cup_{j \leq K-2} \mathcal{F}_j} \mathbb{E}[M_{a,f}(T)] + \tilde{\theta}\mathcal{H}_{K-1} + Z(T,\Delta)\tilde{\theta}|\mathcal{A}_{K-1}|$$

where $(a)$ holds by definition, $(b)$ holds by using $\alpha-$reducilbe structure which ensures that set of superoptimal firms of any agent will lie in markets before it. Next, $(c)$ holds by property of alpha-reducible markets which ensures that for firm $f \in \mathcal{F}_{K-1}$ there exists agent $a' \in \mathcal{A}_{K-1}$ such

that $f = f_{a'}^*$. Next, $(d)$ holds by Lemma 22. Next $(e)$ holds by rearrangement of terms. Next, we show that if (C.8) holds for some $\ell$ then it holds for $\ell + 1$ as well. That is,

$$\tilde{S}_K \underset{(a)}{\leq} \tilde{S}_{K-\ell} + f(\tilde{\theta};\ell)\mathbb{M}_{K-\ell} + \sum_{p=1}^{\ell} g(\tilde{\theta};p) \sum_{a\in\mathcal{A}_{K-p+1}} \sum_{f\in\cup_{j\leq K-\ell-1}\mathcal{F}_j} \mathbb{E}\left[M_{a,f}\right] + \sum_{p=1}^{\ell} f(\tilde{\theta},p)\mathcal{H}_{K-p}$$

$$+ Z(T,\Delta) \sum_{p=1}^{\ell} f(\tilde{\theta},p)|\mathcal{A}_{K-p}|$$

$$\underset{(b)}{=} \tilde{S}_{K-\ell-1} + g(\tilde{\theta};\ell+1)\mathbb{M}_{K-\ell} + \sum_{p=1}^{\ell} g(\tilde{\theta};p) \sum_{a\in\mathcal{A}_{K-p+1}} \sum_{f\in\cup_{j\leq K-\ell-1}\mathcal{F}_j} \mathbb{E}\left[M_{a,f}\right] + \sum_{p=1}^{\ell} f(\tilde{\theta},p)\mathcal{H}_{K-p}$$

$$+ Z(T,\Delta) \sum_{p=1}^{\ell} f(\tilde{\theta},p)|\mathcal{A}_{K-p}|$$

$$\underset{(c)}{=} \tilde{S}_{K-\ell-1} + g(\tilde{\theta};\ell+1)\left(\sum_{a\in\mathcal{A}_{K-\ell}} \sum_{f\in\cup_{j\leq K-\ell-2}\mathcal{F}_j} \mathbb{E}\left[M_{a,f}\right] + \sum_{a\in\mathcal{A}_{K-\ell}} \sum_{f\in\mathcal{F}_{K-\ell-1}} \mathbb{E}\left[M_{a,f}(T)\right]\right)$$

$$+ \sum_{p=1}^{\ell} g(\tilde{\theta};p) \sum_{a\in\mathcal{A}_{K-p+1}} \sum_{f\in\cup_{j\leq K-\ell-1}\mathcal{F}_j} \mathbb{E}\left[M_{a,f}\right] + \sum_{p=1}^{\ell} f(\tilde{\theta},p)\mathcal{H}_{K-p} + Z(T,\Delta) \sum_{p=1}^{\ell} f(\tilde{\theta},p)|\mathcal{A}_{K-p}|$$

$$\underset{(d)}{\leq} \tilde{S}_{K-\ell-1} + g(\tilde{\theta};\ell+1)\left(\sum_{p=1}^{\ell+1} \sum_{a\in\mathcal{A}_{K-p+1}} \sum_{f\in\mathcal{F}_{K-\ell-1}} \mathbb{E}\left[M_{a,f}(T)\right]\right)$$

$$+ \sum_{p=1}^{\ell+1} g(\tilde{\theta};p) \sum_{a\in\mathcal{A}_{K-p+1}} \sum_{f\in\cup_{j\leq K-\ell-2}\mathcal{F}_j} \mathbb{E}\left[M_{a,f}\right] + \sum_{p=1}^{\ell} f(\tilde{\theta},p)\mathcal{H}_{K-p} + Z(T,\Delta) \sum_{p=1}^{\ell} f(\tilde{\theta},p)|\mathcal{A}_{K-p}|$$

$$\underset{(e)}{=} \tilde{S}_{K-\ell-1} + g(\tilde{\theta};\ell+1)\left(\sum_{a'\in\mathcal{A}_{K-\ell-1}} \sum_{p=1}^{\ell+1} \sum_{a\in\mathcal{A}_{K-p+1}} \mathbb{E}\left[M_{a,f_{a'}^*}(T)\right]\right)$$

$$+ \sum_{p=1}^{\ell+1} g(\tilde{\theta};p) \sum_{a\in\mathcal{A}_{K-p+1}} \sum_{f\in\cup_{j\leq K-\ell-2}\mathcal{F}_j} \mathbb{E}\left[M_{a,f}\right] + \sum_{p=1}^{\ell} f(\tilde{\theta},p)\mathcal{H}_{K-p} + Z(T,\Delta) \sum_{p=1}^{\ell} f(\tilde{\theta},p)|\mathcal{A}_{K-p}|$$

$$\underset{(f)}{\leq} \tilde{S}_{K-\ell-1} + g(\tilde{\theta};\ell+1)\left(\tilde{\theta}\mathcal{H}_{K-\ell-1} + \tilde{\theta}\mathbb{M}_{K-\ell-1} + \tilde{\theta}Z(T,\Delta)|\mathcal{A}_{K-\ell-1}|\right)$$

$$+ \sum_{p=1}^{\ell+1} g(\tilde{\theta};p) \sum_{a\in\mathcal{A}_{K-p+1}} \sum_{f\in\cup_{j\leq K-\ell-2}\mathcal{F}_j} \mathbb{E}\left[M_{a,f}\right] + \sum_{p=1}^{\ell} f(\tilde{\theta},p)\mathcal{H}_{K-p} + Z(T,\Delta) \sum_{p=1}^{\ell} f(\tilde{\theta},p)|\mathcal{A}_{K-p}|$$

$$\underset{(g)}{=} \tilde{S}_{K-\ell-1} + f(\tilde{\theta};\ell+1)\mathbb{M}_{K-\ell-1} +$$

$$+ \sum_{p=1}^{\ell+1} g(\tilde{\theta};p) \sum_{a\in\mathcal{A}_{K-p+1}} \sum_{f\in\cup_{j\leq K-\ell-2}\mathcal{F}_j} \mathbb{E}\left[M_{a,f}\right] + \sum_{p=1}^{\ell+1} f(\tilde{\theta},p)\mathcal{H}_{K-p} + Z(T,\Delta) \sum_{p=1}^{\ell+1} f(\tilde{\theta},p)|\mathcal{A}_{K-p}|$$

where $(a)$ is by induction hypothesis, $(b)$ is by decomposing $\tilde{S}_{K-\ell}$, $(c)$ is by using definition of $\mathbb{M}_{K-\ell}$, $(d)$ is by rearrangement of terms and using the fact that $g(\tilde{\theta},\cdot)$ is increasing, $(e)$ is by rearrangement of terms and using the fact that for any $f \in \mathcal{F}_k$ for some $k$ there exists $a' \in \mathcal{A}_k$ such that $f = f_{a'}^*$. Next, $(f)$ is by Lemma 22. Next, $(g)$ is by combining similar terms. This concludes the induction proof.

We know that $\tilde{S}_1 = \mathbb{M}_1 = 0$ because of $\alpha-$reducible structure which ensures that these firms do not have superoptimal firms. Thus in (C.8) if take $\ell = K - 1$ then we get

$$
\tilde{S}_K \leq \sum_{p=1}^{K-1} f(\tilde{\theta}, p)\mathcal{H}_{K-p} + Z(T, \Delta) \sum_{p=1}^{K-1} f(\tilde{\theta}, p)|\mathcal{A}_{K-p}|
$$

$$
\leq \sum_{p=1}^{K-1} \sum_{j=1}^{p} \tilde{\theta}^j \mathcal{H}_{K-p} + Z(T, \Delta) \sum_{p=1}^{K-1} f(\tilde{\theta}, p)|\mathcal{A}_{K-p}|
$$

$$
\leq \sum_{j=1}^{K-1} \tilde{\theta}^j \sum_{p=j}^{K-1} \mathcal{H}_{K-p} + Z(T, \Delta) \sum_{p=1}^{K-1} f(\tilde{\theta}, p)|\mathcal{A}_{K-p}|
$$

$$
\underset{(a)}{=} \sum_{j=1}^{K-1} \tilde{\theta}^j S_{K-j} + Z(T, \Delta) \left(\sum_{j=1}^{K-1} |\mathcal{A}_j|\right) K\tilde{\theta}^{K-1}
$$

$$
\underset{(b)}{\leq} Z(T, \Delta) \left(\sum_{j=1}^{K-1} |\mathcal{A}_j|\right) \sum_{j=1}^{K-1} \tilde{\theta}^j (K - j)\theta^{K-j-1} + Z(T, \Delta) \left(\sum_{j=1}^{K-1} |\mathcal{A}_j|\right) K\tilde{\theta}^{K-1}
$$

where $S_{K-j}$ in (a) is from proof of **(L4)** in Lemma 7 and (b) is by (C.6). Define $\tilde{C}_k = k\tilde{\theta}^{k-1} + \sum_{j=1}^{k-1} \tilde{\theta}^j (k - j)\theta^{k-j-1}$. Thus we see that

$$
\tilde{S}_K \leq |\mathcal{F}| \log(T) \left(1 + \frac{1}{\Delta^2}\right) \left(\sum_{j=1}^{K-1} |\mathcal{A}_j|\right) \tilde{C}_K
$$

$\square$

## D  Proof of Theorem 6

We now look at the joint regret for any $k \in [K]$. Define $Z(T, \Delta) = |F| \log(T) \left(1 + \frac{1}{\Delta^2}\right)$

$$
\sum_{i=1}^{k} \sum_{a \in \mathcal{A}_i} R_a \underset{(a)}{=} \mathcal{O}\Bigg(\sum_{i=1}^{k} \sum_{a \in \mathcal{A}_i} \mathbb{E}[M_{a,\mathbb{F}_a}(T)] + \sum_{i=1}^{k} \sum_{a \in \mathcal{A}_i} \sum_{f \in F\backslash\{f_a^*\}} \mathbb{E}[C_{a,f}(T)] + \sum_{i=1}^{k} \sum_{a \in \mathcal{A}_i} \mathbb{E}[\sum_{t=1}^{T} H_{a,f_a^*}(t)]\Bigg)
$$

$$
\underset{(b)}{=} \mathcal{O}\left(\sum_{i=1}^{k} \sum_{a \in \mathcal{A}_i} \mathbb{E}[M_{a,\underline{\mathbb{F}}_a}(T)] + \sum_{i=1}^{k} \sum_{a \in \mathcal{A}_i} \mathbb{E}[M_{a,\bar{\mathbb{F}}_a}(T)] + \sum_{i=1}^{k} \sum_{a \in \mathcal{A}_i} \mathbb{E}[\sum_{t=1}^{T} H_{a,f_a^*}(t)]\right)
$$

$$
+ \mathcal{O}\left(|\mathcal{F}| \sum_{i=1}^{k} |\mathcal{A}_i| \log(T)\right)
$$

$$
\underset{(c)}{=} \mathcal{O}\left(\sum_{i=1}^{k} \sum_{a \in \mathcal{A}_i} \mathbb{E}[M_{a,\bar{\mathbb{F}}_a}(T)] + \sum_{i=1}^{k} \sum_{a \in \mathcal{A}_i} \mathbb{E}[\sum_{t=1}^{T} H_{a,f_a^*}(t)]\right) + \mathcal{O}(\sum_{i=1}^{k} \sum_{a \in \mathcal{A}_i} |\mathbb{E}_a| Z(T, \Delta))
$$

$$
+ \mathcal{O}\left(|F| \sum_{i=1}^{k} |\mathcal{A}_i| \log(T)\right)
$$

$$
\underset{(d)}{=} \mathcal{O}(\tilde{C}_k \left(\sum_{p=1}^{k} |\mathcal{A}_p|\right) Z(T, \Delta)) + \mathcal{O}(\left(\sum_{p=1}^{k} |\mathcal{A}_p|\right) C_k Z(T, \Delta)) + \mathcal{O}(\sum_{p=1}^{k} \sum_{a \in \mathcal{A}_p} |\mathbb{E}_a| Z(T, \Delta))
$$

$$
+ \mathcal{O}\left(|F| \sum_{p=1}^{k} |\mathcal{A}_p| \log(T)\right)
$$

$$
\underset{(e)}{=} \mathcal{O}\left((C_k + \tilde{C}_k)|\mathcal{F}| \left(\sum_{p=1}^{k} |\mathcal{A}_p|\right)\right) \log(T) \left(1 + \frac{1}{\Delta^2}\right)
$$

where $(a)$ holds due to **(L1)** in Lemma 7, $(b)$ holds due to **(L3)** in Lemma 7, $(c)$ is due to **(L2)** in Lemma 7. Next, $(d)$ is due to **(L4)-(L5)** in Lemma 7. Finally, $(e)$ follows by combining terms.

## E  Technical lemmas

In this section we present some technical lemmas which are helpful in the proofs in next section.

**Lemma 19.** *(Lemma 8.2,Lattimore and Szepesvári (2020)) Let $X_1, X_2, \ldots, X_T$ be a sequence of independent 1-subgaussian random variable, and $\hat{\mu}^{(t)} := \frac{1}{t} \sum_{s=1}^{t} X_s, \epsilon > 0, a > 0$ and*

$$\kappa := \sum_{t=1}^{n} \mathbb{1}\left(\hat{\mu}_t + \sqrt{\frac{2a}{t}} \geq \epsilon\right), \quad \kappa' := u + \sum_{t=\lceil u \rceil}^{T} \mathbb{1}\left(\hat{\mu}_t + \sqrt{\frac{2a}{t}} \geq \epsilon\right)$$

*where $u = \frac{2a}{\epsilon^2}$. Then*

$$\mathbb{E}[\kappa] \leq \mathbb{E}[\kappa'] \leq 1 + \frac{2}{\epsilon^2}(a + \sqrt{\pi a} + 1)$$

**Lemma 20.** *Suppose we use the AB subroutine Algorithm 4 with $\eta \leq 1/50$ then the following two inequalities hold:*

$$\mathbb{E}\left[\sum_{t=1}^{T} \mathbb{1}\left(E_{a,f}^{(r)}(t) = 1, E_{a,f}^{(c)}(t) = 1, H_{a,f}(t)\right)\right] \leq (1+\varpi)\mathbb{E}[M_{a,f}(T)] + \mathcal{O}(\log(T)) + \varpi\mathbb{E}[C_{a,f}(T)],$$
(E.1)

*where $0 < \varpi \leq 32\eta < 1$ and*

$$\mathbb{E}\left[\sum_{t=1}^{T} \mathbb{1}\left(E_{a,f}^{(r)}(t) = 0, E_{a,f}^{(c)}(t) = 1, H_{a,f}^{c}(t)\right)\right] \leq \mathcal{O}\left(\log(T) + \mathbb{E}\left[\sum_{t=1}^{T} \mathbb{1}\left(H_{a,f}(t)\right)\right] + \mathbb{E}[C_{a,f}^{\star}(T)]\right).$$
(E.2)

*Proof.* To simplify the presentation of proof, let's define

$$L_{a,f}^{(\text{adv})}(T) := \sum_{t=1}^{T} \left(\mathbb{1}\left(E_{a,f}^{(r)}(t) = 1, E_{a,f}^{(c)}(t) = 1, H_{a,f}(t)\right) - \mathbb{1}\left(E_{a,f}^{(r)}(t) = 1, E_{a,f}^{(c)}(t) = 1, H_{a,f}^{c}(t)\right)\right)$$

The regret bound for adversarial bandit algorithm from Lemma 11 under $\eta \leq 1/50$ implies

$$\mathbb{E}\left[L_{a,f}^{(\text{adv})}(T)\right] \leq \mathcal{O}(\log(T)) + \varpi\mathbb{E}\left[\min\left\{M_{a,f}^{\star}(T), C_{a,f}^{\star}(T), M_{a,f}(T) + C_{a,f}(T)\right\}\right]$$

$$\mathbb{E}\left[L_{a,f}^{(\text{adv})}(T) - \ell_{a,f}(T)\right] \leq \mathcal{O}(\log(T)) + \varpi\mathbb{E}\left[\min\left\{M_{a,f}^{\star}(T), C_{a,f}^{\star}(T), M_{a,f}(T) + C_{a,f}(T)\right\}\right]$$
(E.3)

where $\varpi \leq 32\eta$ and

$$\ell_{a,f}(T) = \sum_{t=1}^{T} \left(\mathbb{1}\left(E_{a,f}^{(c)}(t) = 1, H_{a,f}(t)\right) - \mathbb{1}\left(E_{a,f}^{(c)}(t) = 1, H_{a,f}^{c}(t)\right)\right)$$

which denotes the total loss received by the adversarial bandit subroutine associated with $(a, f)$ in time $T$ *if* it never take pruning action. Therefore, in (E.3) LHS in first inequality is the regret associated with always pruning. While LHS in second inequality is the regret associated with never pruning.

In the following proof we shall analyze each of the equations in (E.3) separately.

1. The first inequality in (E.3) implies

$$\mathbb{E}\left[\sum_{t=1}^{T} \left(\mathbb{1}\left(E_{a,f}^{(r)}(t) = 1, E_{a,f}^{(c)}(t) = 1, H_{a,f}(t)\right) - \mathbb{1}\left(E_{a,f}^{(r)}(t) = 1, E_{a,f}^{(c)}(t) = 1, H_{a,f}^{c}(t)\right)\right)\right]$$
$$\leq \mathcal{O}(\log(T)) + \varpi\left(\mathbb{E}[M_{a,f}(T) + C_{a,f}(T)]\right).$$

This in turn leads to

$$\mathbb{E}\left[\sum_{t=1}^{T}\left(\mathbb{1}\left(E_{a,f}^{(r)}(t)=1, E_{a,f}^{(c)}(t)=1, H_{a,f}(t)\right)\right)\right] \le \mathbb{E}\left[\mathbb{1}\left(E_{a,f}^{(r)}(t)=1, E_{a,f}^{(c)}(t)=1, H_{a,f}^{\mathsf{c}}(t)\right)\right]$$

$$+ \mathcal{O}(\log(T)) + \frac{1}{2}\left(\mathbb{E}[M_{a,f}(T) + C_{a,f}(T)]\right)$$

$$\le (1+\varpi)\,\mathbb{E}[M_{a,f}(T)] + \mathcal{O}(\log(T)) + \varpi\mathbb{E}[C_{a,f}(T)]$$

2. Using the definition of $\ell_{a,f}(T)$ in the second inequality in (E.3) we obtain

$$\mathbb{E}\left[\sum_{t=1}^{T}\left(-\mathbb{1}\left(E_{a,f}^{(r)}(t)=0, E_{a,f}^{(c)}(t)=1, H_{a,f}(t)\right) + \mathbb{1}\left(E_{a,f}^{(r)}(t)=0, E_{a,f}^{(c)}(t)=1, H_{a,f}^{\mathsf{c}}(t)\right)\right)\right]$$

$$\le \mathcal{O}(\log(T) + \mathbb{E}[\min\{M_{a,f}^{\star}(T), C_{a,f}^{\star}(T)\}])$$

which implies

$$\mathbb{E}\left[\sum_{t=1}^{T}\mathbb{1}\left(E_{a,f}^{(r)}(t)=0, E_{a,f}^{(c)}(t)=1, H_{a,f}^{\mathsf{c}}(t)\right)\right] \le \mathcal{O}\left(\mathbb{E}\left[\sum_{t=1}^{T}\mathbb{1}\left(E_{a,f}^{(r)}(t)=0, E_{a,f}^{(c)}(t)=1, H_{a,f}(t)\right)\right]\right.$$

$$\left. + \mathcal{O}(\log(T)) + \mathbb{E}[\min\{M_{a,f}^{\star}(T), C_{a,f}^{\star}(T)\}]\right)$$

$$\le \mathcal{O}\left(\mathbb{E}\left[\sum_{t=1}^{T}\mathbb{1}\left(H_{a,f}(t)\right)\right] + \log(T) + \mathbb{E}[\min\{M_{a,f}^{\star}(T), C_{a,f}^{\star}(T)\}]\right)$$

This concludes the proof. $\qquad\square$

**Lemma 21** (Pruning stable match). *For any $a \in \mathcal{A}$,*

$$\underbrace{\mathbb{E}\left[\sum_{t=1}^{T}\mathbb{1}\left(E_{a,f_a^*}^{(r)}(t)=0, E_{a,f_a^*}^{(c)}(t)=1\right)\right]}_{\mathbb{E}[\textit{Term I}]} \le \mathcal{O}\left(\mathbb{E}\left[\sum_{t=1}^{T}\mathbb{1}\left(H_{a,f_a^*}(t)\right)\right] + \log(T)\right)$$

*Proof.* We note that

$$\mathbb{E}[\text{Term I}] \le \mathbb{E}\left[\sum_{t=1}^{T}\mathbb{1}\left(E_{a,f_a^*}^{(r)}(t)=0, E_{a,f_a^*}^{(c)}(t)=1, H_{a,f^*}(t)\right) + \sum_{t=1}^{T}\mathbb{1}\left(E_{a,f_a^*}^{(r)}(t)=0, E_{a,f_a^*}^{(c)}(t)=1, H_{a,f^*}^{\mathsf{c}}(t)\right)\right]$$

$$\le \mathcal{O}\left(\mathbb{E}\left[\sum_{t=1}^{T}\mathbb{1}\left(H_{a,f_a^*}(t)\right)\right] + \mathcal{O}(\log(T)) + \mathbb{E}[C_{a,f_a^*}^{\star}(T)]\right)$$

$$\le \mathcal{O}\left(\mathbb{E}\left[\sum_{t=1}^{T}\mathbb{1}\left(H_{a,f_a^*}(t)\right)\right] + \mathcal{O}(\log(T))\right)$$

where the first inequality is due to (E.2) and the last inequality holds due to Lemma 16. $\qquad\square$

**Lemma 22.** *For any $a \in \mathcal{A}$ and $a' \in \mathcal{A}\backslash\{a\}$ we have*

$$\sum_{a' \in \mathcal{A}} \mathbb{E}[M_{a',f_a^*}(T)] \le \mathcal{O}\left(\mathbb{E}\left[\sum_{t=1}^{T}\mathbb{1}\left(H_{a,f_a^*}(t)\right)\right] + |\mathcal{F}|Z(T,\Delta) + \mathbb{E}[M_{a,\bar{\mathbb{F}}_a}(T)]\right)$$

*Proof.* For any agent $a \in \mathcal{A}$ we know that at every time step it either gets matched with some firm or gets collided. This implies

$$\sum_{f' \in \mathcal{F}} \mathbb{E}[C_{a,f'}(T)] + \sum_{f' \in \mathcal{F}\backslash\{f_a^*\}} \mathbb{E}[M_{a,f'}(T)] + \mathbb{E}[M_{a,f_a^*}(T)] = T. \qquad (\text{E.4})$$

Furthermore, in $T$ steps the firm $f_a^*$ can get matched with some agents or remain unmatched. This implies

$$\sum_{a' \in \mathcal{A} \setminus \{a\}} \mathbb{E}[M_{a',f_a^*}(T)] + \mathbb{E}[M_{a,f_a^*}(T)] \leq T. \tag{E.5}$$

Combining (E.4), (E.5) and Lemma 16 we see that

$$\sum_{a' \in \mathcal{A}} \mathbb{E}[M_{a',f_a^*}(T)] \leq \sum_{f' \in \mathcal{F}} \mathbb{E}[C_{a,f'}(T)] + \sum_{f' \in \mathcal{F} \setminus \{f_a^*\}} \mathbb{E}[M_{a,f'}(T)]$$

$$\leq \mathcal{O}\left( \mathbb{E}\left[ \sum_{t=1}^{T} \mathbb{1}\left(H_{a,f_a^*}(t)\right) \right] + |\mathcal{F}|\log(T) \right) + \mathcal{O}\left( \mathbb{E}[M_{a,\underline{\mathbb{F}}_a}(T)] + \mathbb{E}[M_{a,\bar{\mathbb{F}}_a}(T)] \right).$$

Note that from Lemma 15 we have

$$\sum_{a' \in \mathcal{A}} \mathbb{E}[M_{a',f_a^*}(T)] \leq \mathcal{O}\left( \mathbb{E}\left[ \sum_{t=1}^{T} \mathbb{1}\left(H_{a,f_a^*}(t)\right) \right] + |\mathcal{F}|\log(T) + |\underline{\mathbb{F}}_a|Z(T,\Delta) + \mathbb{E}[M_{a,\bar{\mathbb{F}}_a}(T)] \right)$$

$$\leq \mathcal{O}\left( \mathbb{E}\left[ \sum_{t=1}^{T} \mathbb{1}\left(H_{a,f_a^*}(t)\right) \right] + |\mathcal{F}|Z(T,\Delta) + \mathbb{E}[M_{a,\bar{\mathbb{F}}_a}(T)] \right)$$

This completes the proof.

$\square$

# F  Thompson Sampling based Decentralized Matching Algorithm

## F.1  Algorithmic Description

In this section we present a variant of Algorithm 2 but with Thompson sampling based stochastic bandit subroutine. For simplicity, we consider the scenario where the noise in reward is sampled from a normal distribution. To compute the Thompson sampling index each agent $a$ maintains an empirical average of utility generated from any firm $f$ till time $t$ which is $\hat{\mu}_{a,f}(t-1)$. At time step $t$ any agent $a \in \mathcal{A}$ will maintain an index of every firm $f \in \mathcal{F}$ by sampling it from a normal distribution with mean $\hat{\mu}_{a,f}(t-1)$ and variance $\frac{1}{\sum_{f \in \mathcal{F}} M_{a,f}}$ (refer line 3 in Algorithm 5).

## F.2  Bounds for Algorithm 5

We first present the regret bound for Algorithm 5.

**Theorem 23.** *Suppose every agent $a \in \mathcal{A}$ uses Algorithm 5. Then for any $i \in [K]$ :*

$$\sum_{j=1}^{i} \sum_{a \in \mathcal{A}_j} \mathbb{E}[\mathcal{R}_a(T)] = \mathcal{O}\left( C_i |\mathcal{F}||\mathcal{A}| \left( \frac{1}{\Delta^2} \log\left( \frac{1}{\Delta} \right) + \frac{\log(T)}{\Delta^2} + \log(T) \right) \right)$$

*where $\Delta = \min_{a,f} \Delta_{a,f}$ and $C_i$ is a constant dependent on market $\mathcal{M}_i$ and $C_1 < C_2 < ... < C_K$.*

The only difference between proof of Theorem 6 and Theorem 23 is the bound on expected number of matchings with suboptimal firms. We now present the analogue of **(L2)** of Lemma 7 below We now present the lemma before which bounds the number of times agents using Algorithm 5 will get matched to a suboptimal firm:

**Lemma 24.** *For any $i \in [K]$, the expected matches with suboptimal firm satisfies*

$$\sum_{j=1}^{i} \sum_{a \in \mathcal{A}_j} \mathbb{E}[M_{a,\underline{\mathbb{F}}_a}(T)] = \mathcal{O}\left( \sum_{j=1}^{i} \sum_{a \in \mathcal{A}_j} \left( |\underline{\mathbb{F}}_a| \left( \frac{1}{\Delta^2} \log\left( \frac{1}{\Delta} \right) + \frac{\log(T)}{\Delta^2} + \log(T) \right) + \mathbb{E}\left[ \sum_{t=1}^{T} H_{a,f_a^*}(t) \right] \right) \right)$$

*where $\Delta = \min_{a,f} \Delta_a(f)$*

**Algorithm 5:** Thompson Sampling based Decentralized Matching Algorithm (TS-DMA)

---

**Initialize :** $\hat{\mu}_{a,f} = 0, M_{a,f} = 0, p_{a,f} = 0.5, x_{a,f} = 0.5, L_{a,f} = 0, \forall a \in \mathcal{A}, f \in \mathcal{F}$

**1 for** $t = 1, \dots, T$ **do**

**2**     **for** $f \in \mathcal{F}$ **do**

**3**        Sample $\mathcal{T}_{a,f} \sim \mathcal{N}\left(\hat{\mu}_{a,f}, \frac{1}{\bar{M}_a}\right)$, where $\bar{M}_a = \sum_{f \in \mathcal{F}} M_{a,f}$

**4**     **end**

**5**     Set $\mathcal{T}_a = \mathsf{ArgDescendingSort}(\{\mathcal{T}_{a,f}\}_{f \in \mathcal{F}}), i = 1$

**6**     **while** $i \leq n$ **do**

**7**        Set $f = \mathcal{T}_a^{[i]}$

**8**        Sample $P_{a,f} \sim \mathsf{Bernoulli}(p_{a,f})$

**9**        **if** $P_{a,f} = 0$ **then**

**10**          Update $(x_{a,f}, p_{a,f}, L_{a,f}) \longrightarrow \mathsf{AB\_Subroutine}(P_{a,f}, x_{a,f}, p_{a,f}, L_{a,f}, Y_a)$

**11**        **end**

**12**        **if** $P_{a,f} = 1$ **then**

**13**          Query firm $f$

**14**          Receive $(U_a, Y_a)$

**15**          Update $\hat{\mu}_{a,f} \longrightarrow Y_a \frac{\hat{\mu}_{a,f} M_{a,f} + U_a}{M_{a,f} + 1} + (1 - Y_a)\hat{\mu}_{a,f}$ and $M_{a,f} \longrightarrow M_{a,f} + Y_a$,

**16**          Update $(x_{a,f}, p_{a,f}, L_{a,f}) \longrightarrow \mathsf{AB\_Subroutine}(P_{a,f}, x_{a,f}, p_{a,f}, L_{a,f}, Y_a)$

**17**          break while;

**18**        **end**

**19**        $i \longrightarrow i + 1$

**20**     **end**

**21**     **if** $i = |\mathcal{F}| + 1$ **then**

**22**        Query a firm $\mathcal{T}_a^{[1]}$

**23**        Receive $(U_a, Y_a)$

**24**        Update $\hat{\mu}_{a,f} \longrightarrow Y_a \frac{\hat{\mu}_{a,f} M_{a,f} + U_a}{M_{a,f} + 1} + (1 - Y_a)\hat{\mu}_{a,f}$

**25**        Update $M_{a,f} \longrightarrow M_{a,f} + Y_a, \text{ and } \bar{M}_a \longrightarrow \bar{M}_a + Y_a$

**26**     **end**

**27 end**

---

*Proof.* Note that we call an agent $a$ matches with firm $f$ at time $t$ if $Y_a(t) = 1$ and $f_a(t) = f$. Therefore the total number of matchings between $a$ and $f$ till time $T$ is $M_{a,f}(T) = \sum_{t=1}^{T} \mathbb{1}(Y_a(t) = 1, f_a(t) = f)$. Therefore from Lemma 8 the following holds for every $f \in \tilde{\mathcal{F}}$:

$$M_{a,\tilde{\mathcal{F}}}(T) = \sum_{f \in \tilde{\mathcal{F}}} \sum_{t=1}^{T} \mathbb{1}(Y_a(t) = 1, f_a(t) = f)$$

$$\leq \sum_{f \in \tilde{\mathcal{F}}} \sum_{t=1}^{T} \left( \mathbb{1}\left(Y_a(t) = 1, f_a(t) = f, \mathcal{T}_{a,f}(t) \geq \mathcal{T}_{a,f_a^*}(t)\right) + \mathbb{1}\left(E_{a,f}^{(\mathrm{r})}(t) = 1, E_{a,f_a^*}^{(\mathrm{r})} = 0\right) \right)$$

$$\leq \sum_{f \in \tilde{\mathcal{F}}} \sum_{t=1}^{T} \mathbb{1}\left(Y_a(t) = 1, f_a(t) = f, \mathcal{T}_{a,f}(t) \geq \mathcal{T}_{a,f_a^*}(t)\right)$$

$$+ \sum_{t=1}^{T} \sum_{f \in \tilde{\mathcal{F}}} \mathbb{1}\left(E_{a,f}^{(\mathrm{r})}(t) = 1, E_{a,f_a^*}^{(\mathrm{r})} = 0\right)$$

$$\leq \underbrace{\sum_{f \in \tilde{\mathcal{F}}} \sum_{t=1}^{T} \mathbb{1}\left(Y_a(t) = 1, f_a(t) = f, \mathcal{T}_{a,f}(t) \geq \mathcal{T}_{a,f_a^*}(t)\right)}_{\text{Term A}} + \underbrace{\sum_{t=1}^{T} \mathbb{1}\left(E_{a,f_a^*}^{(\mathrm{r})} = 0\right)}_{\text{Term B}}$$

Let's first analyze Term $A$. Define $\mathcal{F}_{t-1} = \{\{f_a(\tau), Y_a(\tau), U_a(\tau)\}_{\tau=1}^{t-1}\}_{a\in\mathcal{A}}$. We first observe that

$$
\mathbb{1}\left(Y_a(t) = 1, E_{a,f}^{(r)}(t) = 1, E_{a,f}^{(c)}(t) = 1, \mathcal{T}_{a,f_a^*} \leq \mathcal{T}_{a,f}(t)\right)
$$

$$
= \underbrace{\mathbb{1}\left(Y_a(t) = 1, E_{a,f}^{(r)}(t) = 1, E_{a,f}^{(c)}(t) = 1, \mathcal{T}_{a,f_a^*} \leq \mathcal{T}_{a,f}(t), \mathcal{T}_{a,f}(t) < \hat{\mu}_{a,f_a^*} - \epsilon\right)}_{\text{Term C}} \tag{F.1}
$$

$$
+ \underbrace{\mathbb{1}\left(Y_a(t) = 1, E_{a,f}^{(r)}(t) = 1, E_{a,f}^{(c)}(t) = 1, \mathcal{T}_{a,f_a^*} \leq \mathcal{T}_{a,f}(t), \mathcal{T}_{a,f}(t) \geq \hat{\mu}_{a,f_a^*} - \epsilon\right)}_{\text{Term D}}
$$

We first provide a bound on Term C. Prior to that let's define some notations. Let's define $G_{a,f}^{(s)}(\epsilon) = 1 - F_{a,f}^{(s)}(\hat{\mu}_{a,f_a^*} - \epsilon)$. Furthermore, conditioned on the event that atleast one arm is pulled, for any agent $a$ let's define $\mathcal{P}_a(t)$ to be the set of firms that are pruned before one is chosen to be played at time $t$. Moreover let $\tilde{A}_{a,f}^{\mathsf{select}}(t)$ be a random variable such that $\tilde{A}_{a,f}^{\mathsf{select}}(t) = 1$ iff $f$ is the firm with maximum index value in all of the non-pruned firms at time $t$. That is, $\tilde{A}_{a,f}^{\mathsf{select}}(t) = \mathbb{1}\left(f \in \arg\max_{f' \in \mathcal{F}\setminus\{\mathcal{P}(t)\cup\{f_a^*\}\}} \mathcal{T}_{a,f'}(t)\right)$. Using this the following holds:

$$
\mathbb{E}[\text{Term C}] = \mathbb{E}[\mathbb{E}[\text{Term C}|\mathcal{F}_{t-1}]]
$$

$$
= \mathbb{E}[\Pr\left(Y_a(t) = 1, E_{a,f}^{(r)}(t) = 1, E_{a,f}^{(c)}(t) = 1, \mathcal{T}_{a,f_a^*} \leq \mathcal{T}_{a,f}(t), \mathcal{T}_{a,f}(t) < \hat{\mu}_{a,f_a^*} - \epsilon|\mathcal{F}_{t-1}\right)]
$$

$$
\leq \mathbb{E}\left[\Pr\left(\mathcal{T}_{a,f_a^*} < \hat{\mu}_{a,f_a^*} - \epsilon|\mathcal{F}_{t-1}\right)\Pr\left(Y_a(t) = 1, \tilde{A}_{a,f}^{\mathsf{select}}(t) = 1, \mathcal{T}_{a,f}(t) < \hat{\mu}_{a,f_a^*} - \epsilon|\mathcal{F}_{t-1}\right)\right] \tag{F.2}
$$

Moreover note that

$$
\Pr\left(Y_a(t) = 1, E_{a,f_a^*}^{(c)}(t) = 1, \mathcal{T}_{a,f}(t)(t) < \hat{\mu}_{a,f_a^*} - \epsilon|\mathcal{F}_{t-1}\right)
$$

$$
\geq \Pr\left(Y_a(t) = 1, \tilde{A}_{a,f}^{\mathsf{select}}(t) = 1, \mathcal{T}_{a,f}(t)(t) < \hat{\mu}_{a,f_a^*} - \epsilon, \mathcal{T}_{a,f_a^*}(t) > \hat{\mu}_{a,f^*} - \epsilon|\mathcal{F}_{t-1}\right)
$$

$$
= \Pr\left(\mathcal{T}_{a,f_a^*}(t) > \hat{\mu}_{a,f_a^*}(t-1) - \epsilon|\mathcal{F}_{t-1}\right)\Pr\left(Y_a(t) = 1, \tilde{A}_{a,f}^{\mathsf{select}}(t) = 1, \mathcal{T}_{a,f}(t)(t) < \hat{\mu}_{a,f_a^*} - \epsilon|\mathcal{F}_{t-1}\right) \tag{F.3}
$$

Using (F.3) in (F.2) we obtain the following

$$
\mathbb{E}[\text{Term C}] = \mathbb{E}\left[\frac{\Pr\left(\mathcal{T}_{a,f_a^*} < \hat{\mu}_{a,f_a^*} - \epsilon|\mathcal{F}_{t-1}\right)}{\Pr\left(\mathcal{T}_{a,f_a^*}(t) > \hat{\mu}_{a,f_a^*}(t-1) - \epsilon|\mathcal{F}_{t-1}\right)}\Pr\left(Y_a(t) = 1, E_{a,f_a^*}^{(c)}(t) = 1, \mathcal{T}_{a,f}(t)(t) < \hat{\mu}_{a,f_a^*} - \epsilon|\mathcal{F}_{t-1}\right)\right]
$$

$$
= \mathbb{E}\left[\frac{1 - G_{a,f_a^*}^{(M_{a,f_a^*}(t-1))}(\epsilon)}{G_{a,f_a^*}^{(M_{a,f_a^*}(t-1))}(\epsilon)}\Pr\left(Y_a(t) = 1, E_{a,f_a^*}^{(c)}(t) = 1, \mathcal{T}_{a,f}(t)(t) < \hat{\mu}_{a,f_a^*} - \epsilon|\mathcal{F}_{t-1}\right)\right]
$$

$$
\leq \mathbb{E}\left[\frac{1 - G_{a,f_a^*}^{(M_{a,f_a^*}(t-1))}(\epsilon)}{G_{a,f_a^*}^{(M_{a,f_a^*}(t-1))}(\epsilon)}\Pr\left(Y_a(t) = 1, E_{a,f_a^*}^{(c)}(t) = 1|\mathcal{F}_{t-1}\right)\right]
$$

Further evaluating the expectation of Term C we have:

$$\mathbb{E}[\text{Term C}] = \sum_{t=1}^{T} \mathbb{E}\left[\frac{1 - G_{a,f_a^*}^{(M_{a,f^*}(t-1))}(\epsilon)}{G_{a,f_a^*}^{(M_{a,f^*}(t-1))}(\epsilon)} \mathbb{1}\left(E_{a,f_a^*}^{(\text{c})}(t) = 1, E_{a,f_a^*}^{(\text{r})}(t) = 1, Y_a(t) = 1\right)\right]$$

$$= \sum_{t=1}^{T}\sum_{s=1}^{t} \mathbb{E}\left[\frac{1 - G_{a,f_a^*}^{(s)}(\epsilon)}{G_{a,f_a^*}^{(s)}(\epsilon)} \mathbb{1}\left(E_{a,f_a^*}^{(\text{c})}(t) = 1, E_{a,f_a^*}^{(\text{r})}(t) = 1, Y_a(t) = 1, M_{a,f_a^*}(t-1) = s\right)\right]$$

$$\leq \mathbb{E}\left[\sum_{s=1}^{T}\frac{1 - G_{a,f_a^*}^{(s)}(\epsilon)}{G_{a,f_a^*}^{(s)}(\epsilon)} \sum_{t=s+1}^{T} \mathbb{1}\left(M_{a,f}(t-1) = s, M_{a,f}(t) = s+1\right)\right]$$

$$\leq \sum_{s=0}^{\infty}\frac{1 - G_{a,f_a^*}^{(s)}(\epsilon)}{G_{a,f_a^*}^{(s)}(\epsilon)}$$

$$\leq \frac{1}{\epsilon^2}\log(\frac{1}{\epsilon})$$

where the last inequality is due to Lattimore and Szepesvári (2020)[13]. Now let's look at Term D. Let's set of time indices when $\mathcal{J}_{a,f} = \{t : G_{a,f}^{(M_{a,f}(t-1))}(\epsilon) > 1/T\}$.

$$\mathbb{E}[\text{Term D}] = \sum_{t=1}^{T} \mathbb{E}\left[\mathbb{1}\left(Y_a(t) = 1, E_{a,f}^{(\text{r})}(t) = 1, E_{a,f}^{(\text{c})}(t) = 1, \mathcal{T}_{a,f_a^*} \leq \mathcal{T}_{a,f}(t), \mathcal{T}_{a,f}(t) \geq \hat{\mu}_{a,f_a^*} - \epsilon\right)\right]$$

$$\leq \underbrace{\sum_{t \in \mathcal{J}_{a,f}} \mathbb{E}\left[\mathbb{1}\left(Y_a(t) = 1, E_{a,f}^{(\text{r})}(t) = 1\right)\right]}_{\text{Term E}} + \underbrace{\sum_{t \notin \mathcal{J}_{a,f}} \mathbb{E}\left[\mathbb{1}\left(\mathcal{T}_{a,f}(t) \geq \hat{\mu}_{a,f_a^*} - \epsilon\right)\right]}_{\text{Term F}}$$

Let's first analyze the Term E above. Note that

$$\sum_{t \in \mathcal{J}_{a,f}} \mathbb{1}\left(Y_a(t) = 1, E_{a,f}^{(\text{r})}(t) = 1\right)$$

$$\leq \sum_{t=1}^{T}\sum_{s=1}^{t-1} \mathbb{1}\left(Y_a(t) = 1, E_{a,f}^{(\text{r})}(t) = 1, G_{a,f}^{s}(\epsilon) > \frac{1}{T}, M_{a,f}(t-1) = s, M_{a,f}(t) = s+1\right)$$

$$= \sum_{s=0}^{T-1} \mathbb{1}\left(G_{a,f}^{(s)}(\epsilon) > \frac{1}{T}\right) \sum_{t=s+1}^{T} \mathbb{1}\left(M_{a,f}(t-1) = s, M_{a,f}(t) = s+1\right)$$

$$= \sum_{s=0}^{T-1} \mathbb{1}\left(G_{a,f}^{(s)}(\epsilon) > \frac{1}{T}\right)$$

$$\leq \mathcal{O}\left(\frac{\log(T)}{(\Delta_{a,f} - \epsilon)^2} + \log(T)\right)$$

where the last property is a property of concentration of normal distribution and is standard in frequentist Thompson sampling analysis. For reader's reference we point to Lattimore and Szepesvári (2020). Next, we bound Term F below:

$$\sum_{t \notin \mathcal{J}_{a,f}} \mathbb{E}\left[\mathbb{1}\left(\mathcal{T}_{a,f}(t) \geq \hat{\mu}_{a,f_a^*} - \epsilon\right)\right] = \sum_{t=1}^{T} \mathbb{E}\left[\mathbb{1}\left(\mathcal{T}_{a,f}(t) \geq \hat{\mu}_{a,f_a^*} - \epsilon, G_{a,f}^{(M_{a,f}(t-1))}(\epsilon) \leq \frac{1}{T}\right)\right]$$

$$= \sum_{t=1}^{T} \mathbb{E}\left[\mathbb{E}\left[\mathbb{1}\left(\mathcal{T}_{a,f}(t) \geq \hat{\mu}_{a,f_a^*} - \epsilon, G_{a,f}^{(M_{a,f}(t-1))}(\epsilon) \leq \frac{1}{T}\right) | \mathcal{F}_{t-1}\right]\right]$$

$$= \sum_{t=1}^{T} \mathbb{E}\left[G_{a,f}^{(M_{a,f}(t-1))}(\epsilon)\mathbb{1}\left(G_{a,f}^{(M_{a,f}(t-1))}(\epsilon) < \frac{1}{T}\right)\right]$$

$$\leq 1$$

---

[13]This inequality is an

Combining the bounds on Term C, Term E and Term F and choosing $\epsilon = \frac{\Delta}{2}$ we have

$$\sum_{f \in \mathbb{F}_a} \mathbb{E}[M_{a,f}(T)] \leq |\mathbb{F}_a| \mathcal{O}\left(\frac{1}{\Delta^2} \log\left(\frac{1}{\Delta}\right) + \frac{\log(T)}{\Delta^2} + \log(T)\right) + \mathbb{E}\left[\sum_{t=1}^{T} \mathbb{1}\left(E_{a,f_a^*}^{(c)}(t) = 1, E_{a,f_a^*}^{(r)}(t) = 0\right)\right]$$

$$\leq |\mathbb{F}_a| \mathcal{O}\left(\frac{1}{\Delta^2} \log\left(\frac{1}{\Delta}\right) + \frac{\log(T)}{\Delta^2} + \log(T)\right) + \mathcal{O}\left(\mathbb{E}\left[\sum_{t=1}^{T} \mathbb{1}\left(H_{a,f_a^*}(t)\right)\right]\right)$$

where the second inequality is due to Lemma 21. This concludes the proof. □

## G  Experimental Study

In this section we present the numerical experiments that demonstrates and validates the results presented in this paper. Moreover, we also observe that our algorithm performs surprisingly well in general market structure, that is in markets which are not $\alpha-$reducible. We leave this as a future work to establish the regret bounds for the proposed algorithms in general markets.

In both sets of experiments, we consider a market comprising of 5 agents and 5 firms. We consider the following two settings:

**(S-I).** randomly initialized preference for agents and randomly initialized (but uniform) preference for firms. This setting ensures that market is $\alpha-$reducible

**(S-II).** randomly initialized preference for agents and firms. In this part we specifically consider setting where $\alpha-$reducibility does *not* hold. This would provide directions for future research in this area.

In our simulations for every agent we randomly sample the preference ordering of firms and assign a mean reward in $[0, 5]$ such that the successful match with most preferred firm gives mean reward 5 and the least preferred firm gives the mean reward 0 and the mean rewards from other firms are equally spaced between $[0, 5]$. The rewards follow a normal distribution with variance 1. We run both Algorithm 2 and Algorithm 5 for 25 times for two randomly sampled preference ordering for each of **(S-I)-(S-II)**.

In Figure 2 we consider **(S-I)** and observe the performance of algorithms. We observe that the mean regret (taken over 25 runs) accumulated by the algorithms saturate very quickly and agents identify their stable match. In Figure 3 we consider **(S-II)** and observe the performance of algorithm. Surprisingly, even without the $\alpha-$reducibility structure, the mean regret[14] (taken over 25 runs) accumulated by the algorithms saturate very quickly and agents identify their stable match. This presents an opportunity to further explore the algorithm presented in this paper for general markets.

Furthermore, in both **(S-I)-(S-II)** we observe that the TS-DMA has higher variance but is faster than UCB-DMA. This is because, compared to UCB-DMA, we observe empirically that TS-DMA very rarely encounters the scenario where all of the firms gets pruned by the adversarial bandit module. We would also like to point that in some cases the regret can be negative (which is desirable) as is shown in Figure 2(c) for the red agent.

---

[14] mean regret here refers to the agent-optimal stable regret(Liu et al., 2021)

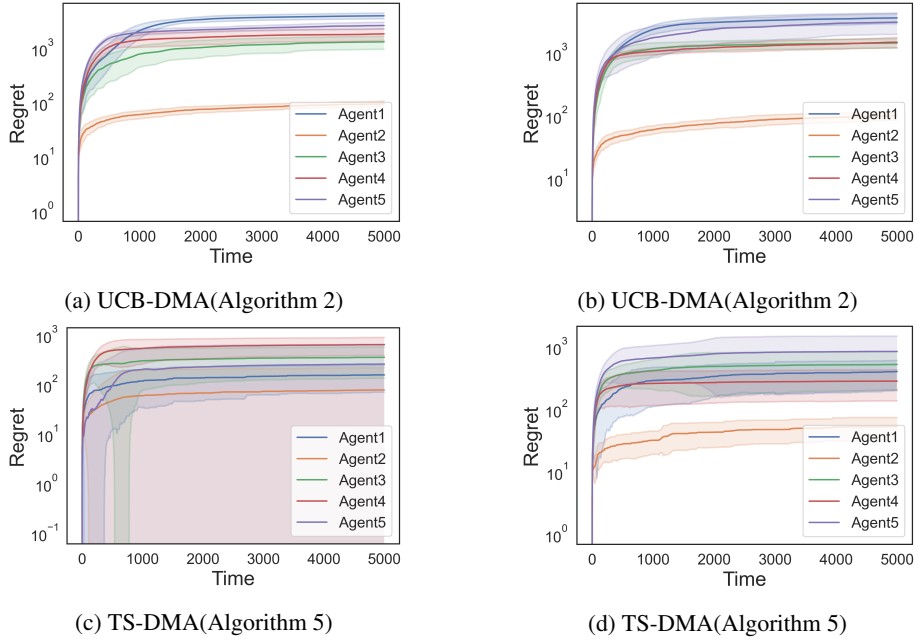

(a) UCB-DMA(Algorithm 2)

(b) UCB-DMA(Algorithm 2)

(c) TS-DMA(Algorithm 5)

(d) TS-DMA(Algorithm 5)

Figure 2: Performance of UCB-DMA (Algorithm 2) and TS-DMA(Algorithm 5) where $\alpha-$reducibilty condition is satisfied. We simulated the algorithm for two randomly generated preference orderings which satisfy the $\alpha$-reducibility condition. The simulation results of one of the preference ordering are presented in left column and for the other in right column. The bold lines and the corresponding shaded region denotes the mean regret and the variance of regret for the agents over 25 runs of the algorithm.

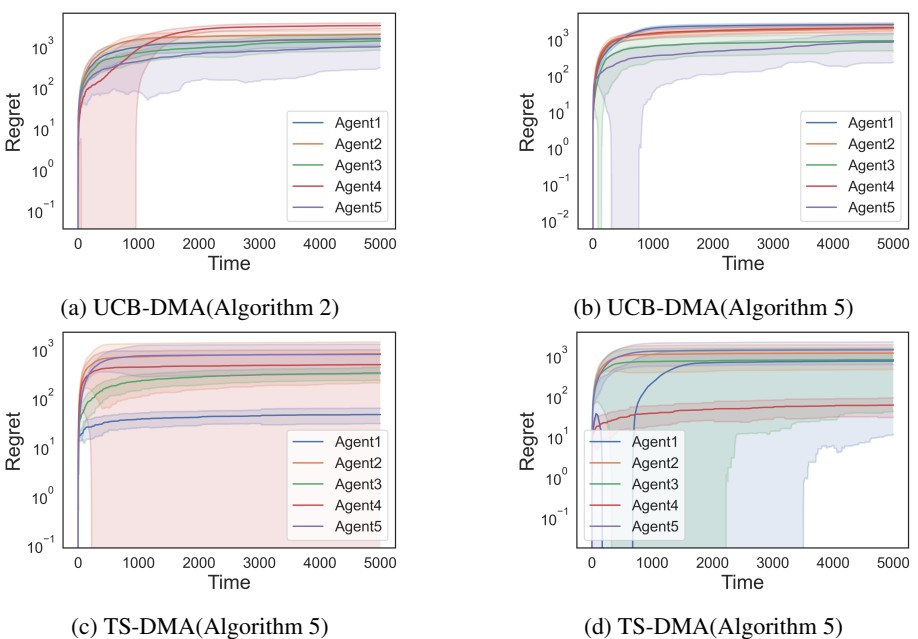

(a) UCB-DMA(Algorithm 2)

(b) UCB-DMA(Algorithm 5)

(c) TS-DMA(Algorithm 5)

(d) TS-DMA(Algorithm 5)

Figure 3: Performance of UCB-DMA (Algorithm 2) and TS-DMA(Algorithm 5) where $\alpha-$reducibilty condition is NOT satisfied. We simulated the algorithm for two randomly generated preference orderings which satisfy the $\alpha$-reducibility condition. The simulation results of one of the preference ordering are presented in left column and for the other in right column. The bold lines and the corresponding shaded region denotes the mean regret and the variance of regret for the agents over 25 runs of the algorithm.

# H Table of Notations

We have accumulated all the main notations used in the paper in form of table below

| Notation | Description |
|----------|-------------|
| $\mathcal{A}$ | Set of agents |
| $\mathcal{F}$ | Set of firms/arms |
| $\mathcal{M}$ | Union of agents and firms |
| $u_a(f)$ | Utility for agent $a$ when matched with firm $f$ |
| $u_f(a)$ | Utility for firm $f$ when matched with agent $a$ |
| $f_a(t)$ | Firm chosen by agent $a$ at time $t$ |
| $f_a^*$ | Stable match of agent $a$ |
| $\overline{\mathbb{F}}_a$ | Set of super-optimal firms for agent $a$ |
| $\underline{\mathbb{F}}_a$ | Set of sub-optimal firms for agent $a$ |
| $K$ | Number of markets formed by decomposition as stated in Remark 3 |
| $\mathcal{A}_i$ | Agents forming fixed pairs after $i-1$ rounds of elimination (Remark 3) |
| $\mathcal{F}_i$ | Firms forming fixed pairs after $i-1$ rounds of elimination (Remark 3) |
| $U_{a,f}$ | Noisy reward that agent $a$ receives on getting matched with firm $f$ |
| $\mathbb{A}_f$ | Set of agents that pull firm $f$ |
| $M_{a,f}(T)$ | Number of times agent $a$ has successfully matched with firm $f$ till time $T$ |
| $C_{a,f}(T)$ | Number of times agent $a$ has collided on firm $f$ till time $T$ |
| $p_{a,f}(t)$ | Probability that agent $a$ will pull firm $f$ at time $t$ |
| $P_{a,f}(t)$ | An indicator if agent $a$ has pulled arm $f$ at time $t$ |
| $Y_a(t)$ | An indicator if agent $a$ got successfully matched at time $t$ |
| $\hat{\mu}_{a,f}(t)$ | Empirical mean of utility derived by agent $a$ on matching with $f$ |
| $\mathcal{I}_{a,f}(t)$ | Index of firm $f$ as computed by agent $a$ at time $t$ |
| $\mathsf{UCB}_{a,f}(t)$ | UCB estimate of reward from firm $f$ to agent $a$ at time $t$ |
| $\mathcal{T}_{a,f}(t)$ | Thompson Sampling index of reward from firm $f$ to agent $a$ at time $t$ |
| $E_{a,f}^{(r)}(t)$ | An indicator if agent $a$ pulled firm $f$ at time $t$ |
| $E_{a,f}^{(c)}(t)$ | An indicator if all the firms with higher index than $f$ got pruned at time $t$ |
| $\tau_{a,f}(T)$ | Time steps during which $E_{a,f}^{(c)}(t) = 1$ |
| $\Delta_{a,f}$ | $u_a(f_a^*) - u_a(f)$ |

Table 1: Table of notations