# OpenReview forum: "Decentralized, Communication- and Coordination-free Learning in Structured Matching Markets"
_NeurIPS.cc/2022/Conference — NeurIPS 2022 Accept_

### Official Review · Reviewer_iWYr · 2022-07-11

**Rating:** 5
**Confidence:** 4
**Soundness:** 3 good
**Presentation:** 3 good
**Contribution:** 3 good

**Summary:**

This paper presents the first framework for constructing decentralized, communication, and coordination-free algorithms for learning while matching. It blends the stochastic bandit algorithm with the adversarial bandit algorithm to solve the statistic problem of learning preference and the competitive problem with other agents respectively. Furthermore, this paper proves an O(C_\alpha |A| |F| \log(T) / (\Delta^2)) regret when the matching market satisfies \alpha-reducible condition.

**Questions:**

1. What’s the regret bound of each single agent? Theorem 5 only provides the regret bound for a summation over a set of agents.

2. Why does this paper study markets under the \alpha-reducible condition, as there are some more general conditions that can guarantee a unique stable matching like SPC and alpha-condition? What makes it hard to analyze the algorithm under those conditions?

3. Could the author explain the constant C_\alpha in more detail? It would be better if the author could give the explicit form of C_alpha and explain how it depends on the market.


**Limitations:**

This paper focuses on the theoretical analysis and there is no potential negative societal impact.

**Strengths And Weaknesses:**

Strength:

1. The contribution of this paper is important since it is the first to propose a decentralized, communication, and coordination-free algorithm for the matching while learning problem.

2. The method that blends the stochastic with adversarial bandit problem is new to handle the learning while matching problem. It gives others an interesting idea to design algorithms when in the face of competition.

3. The submission is technically sound. All claims are well supported by solid theoretical analysis.

4. The structure of the manuscript is well-organized.

Weakness:

1. I think the \alpha-reducible condition, the structure assumption of the matching market in this paper, needs more explanation. It would be better if the author could explain its rationality in more detail, and give some real-life examples that satisfy the \alpha-reducible condition.

2. The constant C_\alpha in the regret bound may grow exponentially in the number of agents, which is a large constant.

3. It would be better if the result could be extended to some weaker conditions such as SPC and \alpha-condition.

4. The experiment lacks some comparisons with existing algorithms. Moreover, it could be better if the author could list the preferences of each agent and firm in the experiment, and explain how this market satisfies \alpha-reducible.

-----
I rechecked the result and thought the exponential term in the regret bound would be an unsatisfying issue.

---

> ### Author Response · Authors · 2022-08-02
> **Response to comments by the reviewer iWYr**
>
> We thank the reviewer for their positive feedback. We respond to each question in the following order:
>
> ### (i) $\alpha$-reducibility condition and relaxation to SPC and alpha-condition:
> We thank the reviewer for prompting us to study SPC and alpha-condition. We note that $\alpha$-reducibility is a stricter condition than SPC and alpha-condition [Clark 2006]. However, the results presented in this work extend directly under the assumption of SPC. This is because all of the results are derived based on the decomposition stated in Remark 3 which can be also obtained from the definition of SPC [Clark 2006, Karpov 2019]. We shall add a remark about the same in the revision. However, as pointed out by Clark 2006, given any population P of agents alpha-reducibility is necessary and sufficient condition for unique stable matching regardless of subpopulation sampled from P. This property does not hold for SPC. Extending the proof with alpha-condition assumption is not a direct extension of current proof technique but we conjecture that the proposed algorithmic paradigm works well in that setting as well. We leave this as an interesting avenue for future research. We shall explicitly state this in the Conclusions section.
>
> [Clark 2006]  The uniqueness of stable matchings.
>
> [Karpov 2019] A necessary and sufficient condition for uniqueness consistency in the stable marriage matching problem
>
>
> ### (ii) Dependence of regret on number of players:
> In the worst case, when the number of submarkets (defined in Remark 3) equals the number of agents, this leads to regret scaling exponentially in the number of agents as we have discussed after Theorem 5. However in the preliminary numerical studies, we find that the regret scales nicely with the number of submarkets. Thus the exponential dependence on the number of submarkets is an artifact of proof technique and it is an interesting direction of future research to improve the dependence of regret bound on the number of players.
>
> ### (iii) Comparison with other algorithms:
> We did not compare the performance of the proposed algorithms with other algorithms as all of the other available algorithms require richer information feedback. We are happy to benchmark against other recent algorithms, but because of the information asymmetry we did not consider the comparison to be fair.
>
> ### (iv) Ensuring alpha-reducibility in experiments:
> In the experiments we considered that firms have uniform preference over the agents which is a sufficient condition for alpha-reducility.
>
> ### (v) Regret bound of single agent:
> Note that from Theorem 5 we can obtain regret bounds for agents lying in different submarkets (defined in Remark 3).
>
> ### (vi) Structure of $C_\alpha$:
> In Theorem 5, for any $k\in [K]$ the term $C^k=\mathcal{O}(k \theta^k)$  where $\theta$ is some constant. When $K=1$ that is there is only one submarket (as per Remark 3) then regret does not depend on the size of the market.

---

> > ### Comment · Reviewer_iWYr · 2022-08-05
> > **Thanks for the reply**
> >
> > Thanks for the explanation about the relationship between the \alpha-reducible and other uniqueness conditions. I think it would be better if you could add this discussion to the revision. You have also solved my confusion about the constant C_{\alpha} in the regret bound, and I also suggest including this discussion in the revision.
> >
> > Since the authors have clarified my doubts, I am happy to increase my score.
> >
> > Best,

---

> > > ### Author Response · Authors · 2022-08-09
> > > **Thanks for your reply**
> > >
> > > We thank reviewer for their reply and increasing the score. We shall definitely incorporate the suggested changes in the final submission.

---

### Official Review · Reviewer_kLiZ · 2022-07-11

**Rating:** 7
**Confidence:** 4
**Soundness:** 3 good
**Presentation:** 2 fair
**Contribution:** 3 good

**Summary:**

This paper studies the bandit learning problem in matching markets. They propose a decentralized, communication- and coordination-free algorithm, which to the best of knowledge is the first such type algorithm in decentralized matching markets. The algorithm consists of both stochastic bandit technique to learn players’ unknown preferences and adversarial bandit technique to deal with the collision among players. They also derive O(log T/\Delta^2) stable regret for markets satisfy \alpha-reducible preferences.

**Questions:**

Please see above weakness.
Coordination:
I have a question on how to understand the ‘coordination-free’? Previous works though need a common communication phase, the only assumption lies in that all agents have common timesteps and common indexes toward arms. So I am wondering that can the proposed algorithm work (or similar results be derived) when different agents come at different round? For example, agent 1 enters in the market in the first round, agent 2 enters in the 100 rounds and so on.


**Limitations:**

The work is about online learning theory and does not have negative societal impact.

**Strengths And Weaknesses:**

Strength:
This paper study a new aspect in online matching markets: whether an algorithm can achieve sublinear stable regret without communication and coordination among players. I think such attempt in the most challenge learning environment is worthwhile.
The algorithm framework is novel in the literature of online matching markets. It looks at players’ learning and conflict from a new perspective different from previous works. It proposes to use stochastic algorithm to deal with the learning and adversarial algorithm to deal with the conflicts. And the proposed algorithm can work with both UCB and TS stochastic algorithms. I think the work is a novel combination of well-known techniques.
Weakness:
The paper should clearly illustrate the alpha-reducible term C in the regret upper bound. During the full paper, I do not see any detailed formula about C. And since C also depends on the market size, it should not be simply regarded as an unimportant constant term.
In line 134, it states that players’ utilities are captured by a real value in R. However, in Line 148, it shows that players will receive 0 reward if it fails the conflict. If the utilities on arms are lower than 0, it seems that the player prefers to be single than being matched with any arm. Is there any error?
Typo:
Algorithm1 line 1: a-> an
Line 3(b) repetitive then

---

> ### Author Response · Authors · 2022-08-02
> **Response to comments by Reviewer kLiZ**
>
> We thank the reviewer for their positive comments. We respond to the comments in the following order:
>
> ### (i) Formula for C:
> In Theorem 5, for any $k\in [K]$ the term $C^k=\mathcal{O}(k \theta^k)$  where $\theta$ is some constant. When $K=1$ that is there is only one submarket  (defined in Remark 3)  then regret does not depend on the size of the market. In the worst case when the number of submarkets is the number of agents then the regret scales exponentially in the number of agents as we have discussed after stating Theorem 5. However in the preliminary numerical studies, we find that the regret scales nicely with the number of submarkets. Thus the exponential dependence on the number of submarkets is an artifact of proof technique and it is an interesting direction of future research to improve the dependence of regret bound on the number of players.
>
> ### (ii) Obtaining negative utility in any round:
> Thank you for catching this, in our analysis we implicitly assume that all utilities are positive. This then implies a form of individual rationality wherein each agent prefers to be matched with a firm rather than collide or not being matched at every round (when viewed in expectation).  At a given round, the realized utility may be negative due to the noise, but we model agents as knowing that successfully matching will— in the long term— give them positive utility.
>
>
> ### (iii) “Meaning of Coordination-free algorithm”:
> The reviewer has correctly mentioned the implication of “coordination free” property. Indeed, the algorithm would seamlessly incorporate an agent  that arrives/departs in between and may alter the preference structures of firms. This can be incorporated by current algorithm as there is no need for any communication protocol like index estimation in [Basu et al 2021] which would require agents to do some “pre-processing” at the global level. Furthermore, existing agents may not even know about the arrival/departure of agents, with time the parameters of stochastic modules and adversarial modules will get appropriately adjusted to incorporate arrival/departures. We shall add a remark about the same in the paper.

---

### Official Review · Reviewer_JTGG · 2022-07-12

**Rating:** 7
**Confidence:** 2
**Soundness:** 3 good
**Presentation:** 3 good
**Contribution:** 3 good

**Summary:**

The paper studies a stable matching problem where the agents have to learn their utilities for the firms. The learning process progresses in rounds. In each round the agents can each propose a firm that they want to be matched to. The firm picks their favorite agent from the set of requests they receive. Agents who are selected by the firms receive a noised measurement of their utility while all the other agents receive nothing. Moreover, the agents act independently and simultaneously in each round. The paper proposes a decentralized algorithm to learn the stable matching and shows that it gives sublinear regrete against the stable matching.

**Questions:**

- How crucial is the assumption of $\alpha$-reducibility? If it does not hold, is there any conjecture whether similar learning outcome can be achieved?

- Can the algorithm be easily extended to the setting where the firms also receives noised utilities?

**Limitations:**

The authors didn't discuss limitations or negative societal impact specifically. However, I don't find any immediate negative societal impact of this work.

**Strengths And Weaknesses:**

Strengths:

- The problem is a natural generalization to the stable matching problem in the full-information setting. The setting is realistic and interesting. The presentation is clear and the paper is easy to follow overall, though at times it requires some domain specific knowledge.

- The results look sound. The algorithm presented is a strong one in that it requires no coordination and communication.

Weakness:

- The subroutine part is a bit dense and the main idea does not stand out as clear as that in the previous section. I didn't check the technical details of this part carefully as a result.

- There are quite some typos. The authors may want to polish the paper a bit more. For example:

    - Line 52: "This setup serves has been..."
    - Line 198: "at time any time..."
    - Algorithm 1, Line 1: "a ordering" --> "an ordering"
    - Line 235: "in the Section 3.1", remove "the"

---

> ### Author Response · Authors · 2022-08-02
> **Response to comments by Reviewer JTGG**
>
> We thank the reviewer for their positive comments and feedback on the presentation. We will incorporate the suggested changes along with improving the presentation before submitting the final manuscript.
>
> The assumption of \alpha-reducibility is required for the proof. However, as per some preliminary numerical experiments we find that the algorithm works equally well even without this assumption. We had included some numerical study on this in the supplementary material (Section G). We shall mention this observation as a potential direction for future research in the “Conclusions” sections.
>
> The scenario where firms are also learning their preferences gets a bit complicated as the firms may incorrectly prefer suboptimal arms which may in turn lead to wrong feedback to arms. This scenario is an interesting avenue for future research.

---

### Official Review · Reviewer_wFQg · 2022-07-12

**Rating:** 7
**Confidence:** 1
**Soundness:** 3 good
**Presentation:** 4 excellent
**Contribution:** 3 good

**Summary:**

This paper studies the problem of online learning in competitive settings in two-sided $\alpha-$reducible matching markets. A general framework for the construction of decentralized, communication, and coordination-free algorithms is presented. It is shown that  the proposed algorithms incurs a regret which grows at most logarithmically in the time horizon under certain conditions.

**Questions:**

No question.

**Limitations:**

Not applicable.

**Strengths And Weaknesses:**

The considered topic is relevant and the key questions that the paper aims to address are well-motivated. The theoretical claims are interesting and appear to be sound. However, I am not knowledgable about this area. Thus, to me, the significance and novelty of this paper are difficult to assess. As for presentation, the paper is well-written and generally easy to follow (even for a non-expert like me).

---

> ### Author Response · Authors · 2022-08-02
> **Response to reviewer wFQg**
>
> We thank reviewer for their positive comments.

---

### Meta-Review · Area_Chair_num2 · 2022-08-25

**Recommendation:** Accept
**Confidence:** Less certain

**Metareview:**

Executive Summary:

The paper studies two-sided matching markets, in the stable matching variety. On one side there are the workers and on the other there are firms. Each worker has a ranking over all firms, and each firm has a ranking over the workers. The goal is to find a stable matching where no two (worker, firm) pairs would prefer to split and re-match.

The paper studies a learning variant of this where the workers initially down know their preferences (which are encoded as non-negative real numbers) but the firms do. Workers can propose to firms. When a firm receives more than one proposal it myopically chooses the worker it likes best. Matched workers learn a noisy version of their value for the firm.

The high-level question the authors ask is: Does there exist a decentralized and coordination-free algorithm that is based only on local history of interactions which provably converges to stable matching?

The main result is affirmative for alpha-reducible markets (Definition 2), which implies the existence of a unique stable matching. It also enables a partition of the market into submarkets (Remark 3). By combining statistical and adversarial learning techniques, they obtain an algorithm in which a worker in submarket M_i experiences a regret of O(C_i  |W| |F| log(T)/ Delta^2 ) against the stable matching. Here W is the set of workers, F is the set of firms, T is the number of rounds, and Delta is the minimum gap in cardinal utility between any two candidates. Crucially, C_k = O(k\theta^k) where k refers to the submarket and \theta is some constant. So when k is of the order of n the "constant" C is actually exponential in n.

The authors conclude that (citing from the abstract) that their main result shows that "competition need not drastically affect the performance of decentralized, communication and coordination free online learning algorithms".

---

Discussion:

Despite the extremely good and unanimously positive scores, I also see very low confidence scores (going as low as "educated guess"). Personally, I am less excited about the paper than the reviewers.

I think what's nice about the paper is that they identify (an established) notion of a structure matching market (alpha-reducibility), and present a decentralized/communication- and coordination-free learning algorithm for it whose regret is parametrized by the number K of submarkets.

A major weakness in my eyes is that the regret bound depends exponentially on the number of submarkets. If I am not mistaken then this means that even for simple markets (such as serial dictatorship markets) the dependence of the regret on the number of agents might be exponential.

There is no discussion of why this form of dependence is necessary.

I still think that this is a nice and non-trivial result, but I feel that the authors slightly overstate the implications (see the last sentence of the abstract), and should better discuss these aspects.

Weak accept.

---

Comments:

** Please specify everywhere (including the abstract and introduction) that the worse case dependency is exponential in the number of AGENTS (and not only the number of sub-markets).

** Please include examples of alpha-reducible markets.

**Award:**

No

---

### Decision · Program_Chairs · 2022-09-14

Accept